# Decentralized Gossip-Based Stochastic Bilevel Optimization over Communication Networks

**Shuoguang Yang**
IEDA, HKUST
yangsg@ust.hk

**Xuezhou Zhang**
Princeton University
xz7392@princeton.edu

**Mengdi Wang**
Princeton University
mengdiw@princeton.edu

## Abstract

Bilevel optimization have gained growing interests, with numerous applications being found in meta learning, minimax games, reinforcement learning, and nested composition optimization. This paper studies the problem of decentralized distributed stochastic bilevel optimization over a network where each agent can only communicate with its neighbors, and gives examples from multi-task, multi-agent learning and federated learning. In this paper, we propose a gossip-based decentralized bilevel learning algorithm that allows networked agents to solve both the inner and outer optimization problems in a single timescale and share information through network propagation. We show that our algorithm enjoys the $\widetilde{\mathcal{O}}(\frac{1}{K\epsilon^2})$ per-agent sample complexity for general nonconvex bilevel optimization and $\widetilde{\mathcal{O}}(\frac{1}{K\epsilon})$ for Polyak-Łojasiewicz objectives, achieving a speedup that scales linearly with the network size $K$. The sample complexities are optimal in both $\epsilon$ and $K$. We test our algorithm on the examples of hyperparameter tuning and decentralized reinforcement learning. Simulated experiments confirmed that our algorithm achieves the state-of-the-art training efficiency and test accuracy.

## 1 Introduction

In recent years, stochastic bilevel optimization (SBO) has attracted increasing attention from the machine learning community. It has been found to provide favorable solutions to a variety of problems, such as meta learning and hyperparameter optimization (Franceschi et al., 2018; Snell et al., 2017; Bertinetto et al., 2018), composition optimization (Wang and Liu, 2016), two-player games (Von Stackelberg and Von, 1952), reinforcement learning and imitation learning (Arora et al., 2020; Hong et al., 2020). While the majority of the above mentioned work focuses on algorithm designs in the classic centralized setting, such problems often arise in distributed/federated applications, where agents are unwilling to share data but rather perform local updates and communicate with neighbors. Theories and algorithms for distributed stochastic bilevel optimization are less developed.

Consider the *decentralized* learning setting where the data are distributed over $K$ agents $\mathcal{K} = \{1, 2, \cdots, K\}$ over a communication network. Each agent can only communicate with its neighbors over the network. One example is federated learning which is often concerned with a single-server-multi-user system, where agents communicate with a central server to solve a task cooperatively (Lan and Zhou, 2018; Ge et al., 2018). Another example is the sensor network, where sensors are fully decentralized and can only communicate with nearby neighbors (Taj and Cavallaro, 2011).

We consider the following decentralized stochastic bilevel optimization (DSBO) problem,

$$\min_{x \in \mathbb{R}^{d_x}} F(x) = \left\{ \frac{1}{K} \sum_{k=1}^{K} f^k(x, y^\star(x)) \right\}, \text{ s.t. } y^\star(x) = \operatorname*{argmin}_{y \in \mathbb{R}^{d_y}} \left\{ \frac{1}{K} \sum_{k=1}^{K} g^k(x, y) \right\}, \quad (1)$$

36th Conference on Neural Information Processing Systems (NeurIPS 2022).

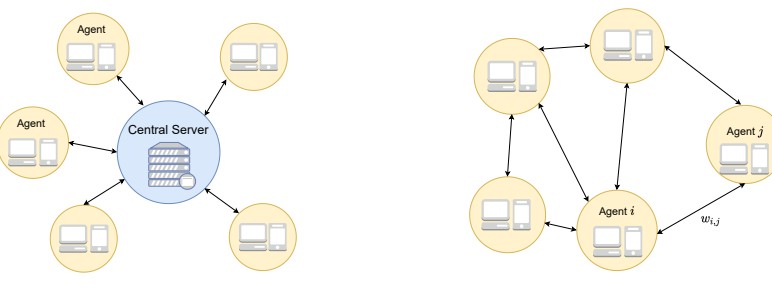

(a) Single-Server-Multi-Agent Network      (b) Decentralized Network

Figure 1: Illustration of Distributed Network Structures. In a centralized network, the agents may cooperate to solve a specific task by communicating with the central server, commonly seen in federated learning.

where $f^k(x, y) = \mathbb{E}_{\zeta^k}[f^k(x, y; \zeta^k)]$ and $g^k(x, y) = \mathbb{E}_{\xi^k}[g^k(x, y; \xi^k)]$ may vary between agents. The expectations $\mathbb{E}_{\zeta^k}[\bullet]$ and $\mathbb{E}_{\xi^k}[\bullet]$ are taken with respect to the random variables $\zeta^k$ and $\xi^k$, with heterogeneous distributions across agents. We consider the scenario where each $g^k(x, y)$ is strongly convex in $y$. We use the notation $F^* = \min_{x \in \mathbb{R}^{d_x}} F(x)$, $f(x, y) = \frac{1}{K} \sum_{k \in \mathcal{K}} f^k(x, y)$, and $g(x, y) = \frac{1}{K} \sum_{k \in \mathcal{K}} g^k(x, y)$ for convenience.

## 1.1 Example applications of SBO

SBO was first employed to formulate the resource allocation problem (Bracken and McGill, 1973) and has since found its applications in many classic operations research settings (Cramer et al., 1994; Sobieszczanski-Sobieski and Haftka, 1997; Livne, 1999; Tu et al., 2020; Tu, 2021), and more recently in machine learning problems (Franceschi et al., 2018; Snell et al., 2017; Bertinetto et al., 2018; Wang and Liu, 2016). In particular, we introduce two applications that have recently attracted lot of attention, namely hyperparameter optimization and compositional optimization.

**Hyperparameter Optimization** The problem of hyper-parameter tuning (Okuno et al., 2021) often takes the following form:

$$\min_{x \in \mathbb{R}^d} \left\{ \sum_{i \in \mathcal{D}_{\text{val}}} \ell_i(y^\star(x)) \right\}, \text{ s.t. } y^\star(x) \in \operatorname*{argmin}_{y \in \mathbb{R}^{d_y}} \left\{ \sum_{j \in \mathcal{D}_{\text{train}}} \ell_j(y) + \mathcal{R}(x, y) \right\}, \quad (2)$$

where $\mathcal{D}_{\text{train}}$ and $\mathcal{D}_{\text{val}}$ are two datasets used for training and validation, respectively, $y \in \mathbb{R}^{d_y}$ is a vector of unknown parameters to optimize, $\ell_j(y)$ is a convex loss over data $i$, and $x \in \mathbb{R}^d$ is a vector of hyper-parameters for a strongly convex regularizer $\mathcal{R}(x, y)$. For any hyper-parameter $x \in \mathbb{R}^d$, the inner-level problem solves for the best parameter $y^\star(x)$ over the training set $\mathcal{D}_{\text{train}}$ under the regularized training loss $\ell_j(y) + \mathcal{R}(x, y)$. The goal is to find the hyper-parameter $x^* \in \mathbb{R}^d$ whose corresponding best response $y^\star(x)$ yields the least loss over the validation set $\mathcal{D}_{\text{val}}$. In practice, continuous hyperparameters are often tuned by a grid search which is exponentially expensive. An efficient SBO algorithm should find the optimal parameters in time increasingly polynomially with the dimension, rather than exponentially. When the training and validation sets are distributed across nodes, the problem becomes a distributed SBO.

**Compositional Optimization** Let $g(x, \xi) : \mathbb{R}^d \to \mathbb{R}^{d_y}$ and $f(y, \zeta) : \mathbb{R}^{d_y} \to \mathbb{R}$ be two stochastic mappings. Stochastic compositional optimization (SCO) (Wang et al., 2017a) takes the form

$$\min_{x \in \mathcal{X}} \mathbb{E}_\zeta[f(\mathbb{E}_\xi[h(x; \xi)]; \zeta)].$$

SCO applies to risk management (Yang et al., 2019; Ruszczynski, 2021), machine learning (Chen et al., 2021b), and reinforcement learning (Wang et al., 2017b). SCO was identified as a special case of SBO by (Chen et al., 2021c). To see this, we take the inner optimization objective to be $g(x, y) = \mathbb{E}_\xi[(y - h(x, \xi))^2]$. Thus $y^\star(x) = \operatorname{argmin}_y g(x, y)$ becomes $y^\star(x) = \mathbb{E}_\xi[h(x; \xi)]$, arriving at an instance of SBO.

## 1.2 Challenges with Distributed SBO

Despite the recent rapid development of distributed single-level optimization, a method appropriate to DSBO remains elusive. The major hurdle to solving DSBO lies in the absence of explicit knowledge of $y^\star(x)$, so that an unbiased gradient for $\nabla f(x, y^\star(x))$ is not available. Recently, by applying the implicit function theorem, (Couellan and Wang, 2016; Ghadimi and Wang, 2018) showed that the gradient of a non-distributed SBO can be expressed as

$$\nabla_x f(x, y^\star(x)) - \nabla^2_{xy} g(x, y^\star(x))[\nabla^2_{yy} g(x, y^\star(x))]^{-1} \nabla_y f(x, y^\star(x)), \tag{3}$$

providing a connection between SBO and classical stochastic optimization. Since then, various algorithms have been proposed to obtain sharp estimators for $y^\star(x)$ and reduce the bias of the constructed gradients (Chen et al., 2021a; Hong et al., 2020; Ji et al., 2021; Yang et al., 2021). These techniques result in tight convergence analysis and give rise to algorithms widely used in applications. However, no prior algorithms can be applied to the distributed setting.

Solving Problem (1) becomes challenging in the distributed setting in several aspects:

1. Even in single-agent SBO, the lack of $y^\star(x)$ makes outer optimization nontrivial, and estimating $y^\star(x)$ requires additional stochastic approximation, weighted averaging, and sophisticated calculation.

2. Calculating the outer gradient is highly nontrivial, even when we have an inner solution $y$. Note that

$$\frac{1}{K} \sum_{k \in \mathcal{K}} \left( \nabla^2_{xy} g^k(x, y)[\nabla^2_{yy} g^k(x, y)]^{-1} \nabla_y f^k(x, y) \right) \neq \nabla^2_{xy} g(x, y)[\nabla^2_{yy} g(x, y)]^{-1} \nabla_y f(x, y). \tag{4}$$

   In other words, even if the inner problem is solved, the outer gradient requires a new estimation mechanism.

3. Now we move to SBO over distributed networks. In network learning, communication between agents can be limited by the network structure and communication protocol, so taking a simple average across agents may require multiple communication rounds.

Because of the above difficulties, it remains unclear how to estimate the outer gradient sharply under a distributed network. In an attempt to tackle this problem, this paper studies the convergence theory and sample complexity of gossip-based algorithms. In particular, we ask two theoretical questions:

*(i) How does the sample complexity of DSBO scale with the optimality gap and network size?*
*(ii) How is the efficiency of DSBO affected by the network structure?*

In this paper, we develop a gossip-based stochastic approximation scheme where each agent solves an optimization problem collaboratively by sampling stochastic first- and second-order information using its data and making gossip communications with its neighbors. In addition, we develop novel techniques for convergence analysis to characterize the convergence behavior of our algorithm. To the best of our knowledge, our work is the first to formulate DSBO mathematically and propose an algorithm with theoretical convergence guarantees. Moreover, we show that our algorithm enjoys an $\widetilde{\mathcal{O}}(\frac{1}{K\epsilon^2})$ sample complexity for finding $\epsilon$-stationary points for nonconvex objectives, where $\widetilde{\mathcal{O}}(\cdot)$ hides logarithmic factors, and enjoys an $\widetilde{\mathcal{O}}(\frac{1}{K\epsilon})$ sample complexity for Polyak-Łojasiewicz (PL) functions, subsuming strongly convex optimization. These results subsume the state-of-the-art results for non-federated stochastic bilevel optimization (Chen et al., 2021c) and central-server regimes (Tarzanagh et al., 2022), showing that almost no degradation is induced by network consensus. Further, the above results suggest that our algorithm exhibits a linear speed-up effect for decentralized settings; that is, the required per-agent sample complexities decrease linearly with the number of agents.

## 2 Related Works

Bilevel optimization was first formulated by (Bracken and McGill, 1973) for solving resource allocation problems. Later, a class of constrained-based algorithms was proposed by (Hansen et al., 1992; Shi et al., 2005), which treats the inner-level optimality condition as constraints to the out-level problem. Recently, (Couellan and Wang, 2016) examined the finite-sum case for unconstrained strongly convex lower-lower problems and proposed a gradient-based algorithm that

| Stochastic Bilevel Optimization | | | | | | |
|---|---|---|---|---|---|---|
| | BSA | stocBio | ALSET | FEDNEST | This Work | |
| Network | Single-Agent | | | Central-Server | Decentralized | |
| Samples in $\zeta$ | $\mathcal{O}(\epsilon^{-2})$ | $\mathcal{O}(\epsilon^{-2})$ | $\mathcal{O}(\epsilon^{-2})$ | $\mathcal{O}(\epsilon^{-2})$ | $\mathcal{O}(\frac{1}{K\epsilon^2})$ | $\mathcal{O}(\frac{1}{K\epsilon^2} + \frac{K}{(1-\rho)^2\epsilon})$ |
| Samples in $\xi$ | $\widetilde{\mathcal{O}}(\epsilon^{-3})$ | $\widetilde{\mathcal{O}}(\epsilon^{-2})$ | $\widetilde{\mathcal{O}}(\epsilon^{-2})$ | $\widetilde{\mathcal{O}}(\epsilon^{-2})$ | $\widetilde{\mathcal{O}}(\frac{1}{K\epsilon^2})$ | $\widetilde{\mathcal{O}}(\frac{1}{K\epsilon^2} + \frac{K}{(1-\rho)^2\epsilon})$ |

| Stochastic Compositional Optimization | | | | | | |
|---|---|---|---|---|---|---|
| | SCGD | NASA | ALSET | FEDNEST | This Work | |
| Network | Single-Agent | | | Central-Server | Decentralized | |
| Samples in $\zeta$ | $\mathcal{O}(\epsilon^{-4})$ | $\mathcal{O}(\epsilon^{-2})$ | $\mathcal{O}(\epsilon^{-2})$ | $\mathcal{O}(\epsilon^{-2})$ | $\mathcal{O}(\frac{1}{K\epsilon^2})$ | $\mathcal{O}(\frac{1}{K\epsilon^2} + \frac{K}{(1-\rho)^2\epsilon})$ |
| Samples in $\xi$ | $\mathcal{O}(\epsilon^{-4})$ | $\mathcal{O}(\epsilon^{-2})$ | $\mathcal{O}(\epsilon^{-2})$ | $\mathcal{O}(\epsilon^{-2})$ | $\mathcal{O}(\frac{1}{K\epsilon^2})$ | $\mathcal{O}(\frac{1}{K\epsilon^2} + \frac{K}{(1-\rho)^2\epsilon})$ |

Table 1: Summary of per-agent sample complexities for nonconvex stochastic bilevel and compositional optimization in different settings: BSA (Ghadimi and Wang, 2018), ALSET (Chen et al., 2021c), stocBio (Ji et al., 2021), FEDNEST (Tarzanagh et al., 2022), SCGD (Wang and Liu, 2016), and NASA (Ghadimi et al., 2020). Given the weight matrix $W$, $\rho = \|W - \frac{1}{K}\mathbf{1}\mathbf{1}^\top\|_2^2$.

exhibits asymptotic convergence under certain step-sizes. For SBO, (Ghadimi and Wang, 2018) developed a double-loop algorithm and established the first known complexity results. Subsequently, various methods have been employed to improve the sample complexity, including two-timescale stochastic approximation (Hong et al., 2020), acceleration (Chen et al., 2021a), momentum (Khanduri et al., 2021), and variance reduction (Guo et al., 2021; Ji et al., 2021; Yang et al., 2021).

Distributed optimization was developed to handle real-world large-scale datasets (Dekel et al., 2012; Feyzmahdavian et al., 2016) and graph estimation (Wang et al., 2015). Centralized and decentralized systems are both important problems that have drawn significant attention. A centralized system considers the network topology where there is a central agent that communicates with the remaining agents (Lan and Zhou, 2018), while in a decentralized system (Gao and Huang, 2020; Koloskova et al., 2019; Lan et al., 2020; McMahan et al., 2017), each agent can only communicate with its neighbors by using gossip (Lian et al., 2017b) or gradient tracking (Pu and Nedić, 2021) communication strategies, with applications in multi-agent reinforcement learning (Xu et al., 2020). Variance reduction approaches (Xin et al., 2020, 2021; Lian et al., 2017a) have also been applied to improve the convergence rate of decentralized optimization. Random projection schemes have been studied to handle large sets of constraints (Wang and Bertsekas, 2015, 2016; Liu et al., 2015). All of the above trials were made on vanilla stochastic optimization problems. We are the first to study the decentralized bilevel optimization problem to our knowledge. Table 1 compares this work with prior arts under different settings.

## 3   Problem Setup

**Assumption 3.1 (Sampling Oracle $\mathcal{SO}$)** *Agent $k$ may query the sampler and receive an independent locally sampled unbiased first- and second-order information $\nabla_x f^k(x, y; \zeta^k)$, $\nabla_y f^k(x, y; \xi^k)$, $\nabla_y g^k(x, y; \xi^k)$, $\nabla_{yy}^2 g^k(x, y; \xi^k)$, and $\nabla_{xy}^2 g^k(x, y; \xi^k)$.*

**Assumption 3.2 (Gossip Protocol)** *The network gossip protocol is specified by a $K \times K$ symmetric weight matrix $W$ with nonnegative entries. Each agent $k$ may receive information from its neighbors, e.g., $z_j, j \in \mathcal{N}_k$, and aggregate them by a weighted sum $\sum_{j \in \mathcal{N}_k} w_{k,j} z_j$. Further, matrix $W$ satisfies*

   *(i) $W$ is doubly stochastic such that $\sum_i w_{i,j} = 1$ and $\sum_j w_{i,j} = 1$ for all $i, j \in [K]$.*

   *(ii) There exists a constant $\rho \in (0, 1)$ such that $\|W - \frac{1}{K}\mathbf{1}\mathbf{1}^\top\|_2^2 = \rho$, where $\|A\|_2$ denotes the spectral norm of $A \in \mathbb{R}^{K \times K}$.*

These assumptions on the adjacency matrix are crucial to ensure the convergence of decentralized algorithms and are commonly made in the literature of decentralized optimization (Lian et al., 2017b).

**Assumption 3.3** *Let $C_f, L_f$ be positive scalars. The outer level functions $\{f^k\}_{k \in \mathcal{K}}$ satisfy the followings.*

*(i)* *There exists at least one optimal solution to Prob. (1).*

*(ii)* *Both $\nabla_x f^k(x, y)$ and $\nabla_y f^k(x, y)$ are $L_f$-Lipschitz continuous in $(x, y)$ such that for all $x_1, x_2 \in \mathbb{R}^{d_x}$ and $y_1, y_2 \in \mathbb{R}^{d_y}$,*

$$\|\nabla_x f^k(x_1, y_1) - \nabla_x f^k(x_2, y_2)\| \leq L_f(\|x_1 - x_2\| + \|y_1 - y_2\|),$$
$$\text{and } \|\nabla_y f^k(x_1, y_1) - \nabla_y f^k(x_2, y_2)\| \leq L_f(\|x_1 - x_2\| + \|y_1 - y_2\|).$$

*(iii)* *For all $x \in \mathbb{R}^{d_x}$ and $y \in \mathbb{R}^{d_y}$,*

$$\mathbb{E}[\|\nabla_y f^k(x, y; \zeta^k)\|^2] \leq C_f^2 \text{ and } \mathbb{E}[\|\nabla_x f^k(x, y; \zeta^k)\|^2] \leq C_f^2.$$

**Assumption 3.4** *Let $C_g, L_g, \widetilde{L}_g, \mu_g, \kappa_g$ be positive scalars. The inner level functions $\{g^k\}_{k \in \mathcal{K}}$ satisfy the followings.*

*(i)* *For all $x \in \mathbb{R}^{d_x}$, $g(x, y)$ is $\mu_g$-strongly convex in $y$.*

*(ii)* *For all $x \in \mathbb{R}^{d_x}$ and $y \in \mathbb{R}^{d_y}$, $g^k(x, y)$ is twice continuously differentiable in $(x, y)$.*

*(iii)* *$\nabla_y g^k(x, y)$, $\nabla^2_{xy} g^k(x, y)$, and $\nabla^2_{yy} g^k(x, y)$ are Lipschitz continuous in $(x, y)$ such that for all $x_1, x_2 \in \mathbb{R}^{d_x}$ and $y_1, y_2 \in \mathbb{R}^{d_y}$,*

$$\|\nabla_y g^k(x_1, y_1) - \nabla_y g^k(x_2, y_2)\| \leq L_g(\|x_1 - x_2\| + \|y_1 - y_2\|),$$
$$\|\nabla^2_{xy} g^k(x_1, y_1) - \nabla^2_{xy} g^k(x_2, y_2)\|_F \leq \widetilde{L}_g(\|x_1 - x_2\| + \|y_1 - y_2\|),$$
$$\text{and } \|\nabla^2_{yy} g^k(x_1, y_1) - \nabla^2_{yy} g^k(x_2, y_2)\|_F \leq \widetilde{L}_g(\|x_1 - x_2\| + \|y_1 - y_2\|).$$

*(iv)* *For all $x \in \mathbb{R}^{d_x}, y \in \mathbb{R}^{d_y}$, $\nabla_y g^k(x, y; \xi^k)$, $\nabla^2_{yy} g^k(x, y; \xi^k)$, and $\nabla^2_{xy} g^k(x, y; \xi^k)$ have bounded second-order moments such that $\mathbb{E}[\|\nabla_y g^k(x, y; \xi^k)\|^2] \leq C_g^2$, $\mathbb{E}[\|\nabla^2_{yy} g^k(x, y; \xi^k)\|_2^2] \leq L_g^2$, and $\mathbb{E}[\|\nabla^2_{xy} g^k(x, y; \xi^k)\|_2^2] \leq L_g^2$.*

*(v)* *For all $x \in \mathbb{R}^{d_x}, y \in \mathbb{R}^{d_y}$, $\mathbf{I} - \frac{1}{L_g} \nabla^2_{yy} g^k(x, y; \xi^k)$ has bounded second moment such that $\mathbb{E}[\|\mathbf{I} - \frac{1}{L_g} \nabla^2_{yy} g^k(x, y; \xi^k)\|_2^2] \leq (1 - \kappa_g)^2$, where $0 < \kappa_g \leq \frac{\mu_g}{L_g} \leq 1$.*

We defer the detailed assumptions to Section A.1 of the supplement.

Note that we denote by $\|A\|_2 = \sigma_{\max}(A)$ the induced 2-norm for any matrix $A$. Here we point out that the above assumptions allow heterogeneity between functions $f^k$'s and $g^k$'s over the agents, and the smoothness and boundedness conditions are commonly adopted in SBO (Hong et al., 2020; Chen et al., 2021a).

## 4 Algorithm

As discussed in Section 1.2, the key challenge to solving DSBO is that each agent only has access to its own data but is required to construct estimators for the gradients and Hessian averaged across all agents. It is particularly challenging to construct such estimators when limited by the network's communication protocol.

To overcome this issue, we propose a gossip-based DSBO algorithm where each agent $k \in \mathcal{K}$ iteratively updates a solution pair $(x_t^k, y_t^k)$ by using the combination of gossip communications and weighted-average stochastic approximation, where $y_t^k$ serves as an estimator of the best response $y_t^* := y^*(\bar{x}_t)$ with $\bar{x}_t := \frac{1}{K} \sum_{k \in \mathcal{K}} x_t^k$. The full algorithm is given in Alg. 1.

Here we briefly explain the idea of our DSBO algorithm. Suppose agent $k$ would like to estimate $\nabla_x f(x_t^k, y_t^k)$ by $s_t^k$, under Alg. 1 Step 6, it would query the stochastic first-order information using its own data, make gossip communications with neighbors, and update its estimators by taking the weighted average of its previous estimate $s_{t-1}^k$, neighbors $j$'s estimate $s_{t-1}^j$, and the newly sampled gradient $\nabla_x f^k(x_t^k, y_t^k; \zeta_t^k)$. Roughly speaking, this procedure can be viewed as taking a weighted average of the gradients sampled by all agents over the network, except that the effect of consensus should also been taken into account.

---

**Algorithm 1** Gossip-Based Decentralized Stochastic Bilevel Optimization

---

**input:** Step-sizes $\{\alpha_t\}$, $\{\beta_t\}$, $\{\gamma_t\}$, total iterations $T$, sampling oracle $\mathcal{SO}$, weight matrix $W$, smoothness constant $L_g$, hessian sampling parameter $b$

$\quad x_0^k = \mathbf{0}, y_0^k = \mathbf{0}, u_0^k = \mathbf{0}, s_0^k = \mathbf{0}, h_0^t = \mathbf{0}, v_{0,i}^k = \mu_g \mathbf{I}$ for $i = 1, 2, \cdots, b$

1: **for** $t = 0, 1, \cdots, T - 1$ **do**
2: $\quad$ **for** $k = 1, \cdots, K$ **do**
3: $\quad\quad$ Local sampling: Query $\mathcal{SO}$ at $(x_t^k, y_t^k)$ to obtain $\nabla_x f^k(x_t^k, y_t^k; \zeta_t^k)$, $\nabla_y f^k(x_t^k, y_t^k; \zeta_t^k)$,
$\quad\quad\quad\quad \nabla_y g^k(x_t^k, y_t^k; \xi_t^k), \nabla_{xy}^2 g^k(x_t^k, y_t^k; \xi_t^k)$, and $\{\nabla_{yy}^2 g^k(x_t^k, y_t^k; \xi_{t,i}^k)\}_{i=1}^b$.
4: $\quad\quad$ Outer loop update: $x_{t+1}^k = \sum_{j \in \mathcal{N}_k} w_{k,j} x_t^j - \alpha_t \left( s_t^k - u_t^k q_t^k h_t^k \right)$.
5: $\quad\quad$ Inner loop update: $y_{t+1}^k = \sum_{j \in \mathcal{N}_k} w_{k,j} y_t^j - \gamma_t \nabla_y g^k(x_t^k, y_t^k; \xi_t^k)$.
6: $\quad\quad$ Estimate $\nabla_x f(x_t, y_t)$: $s_{t+1}^k = (1 - \beta_t) \sum_{j \in \mathcal{N}_k} w_{k,j} s_t^j + \beta_t \nabla_x f^k(x_t^k, y_t^k; \zeta_t^k)$.
7: $\quad\quad$ Estimate $\nabla_y f(x_t, y_t)$: $h_{t+1}^k = (1 - \beta_t) \sum_{j \in \mathcal{N}_k} w_{k,j} h_t^j + \beta_t \nabla_y f^k(x_t^k, y_t^k; \zeta_t^k)$.
8: $\quad\quad$ Estimate $\nabla_{xy}^2 g(x_t, y_t)$: $u_{t+1}^k = (1 - \beta_t) \sum_{j \in \mathcal{N}_k} w_{k,j} u_t^j + \beta_t \nabla_{xy}^2 g^k(x_t^k, y_t^k; \xi_t^k)$.
9: $\quad\quad$ Estimate $[\nabla_{yy}^2 g(x_t, y_t)]^{-1}$: Set $Q_{t+1,0}^k = \mathbf{I}$
10: $\quad\quad$ **for** $i = 1, \cdots, b$ **do**
11: $\quad\quad\quad$ $v_{t+1,i}^k = (1 - \beta_t) \sum_{j \in \mathcal{N}_k} w_{k,j} v_{t,i}^j + \beta_t \nabla_{yy}^2 g^k(x_t^k, y_t^k; \xi_{t,i}^k)$,
12: $\quad\quad\quad$ $Q_{t+1,i}^k = \mathbf{I} + (\mathbf{I} - \frac{1}{L_g} v_{t+1,i}^k) Q_{t+1,i-1}^k$
13: $\quad\quad$ **end for**
14: $\quad\quad$ Set $q_{t+1}^k = \frac{1}{L_g} Q_{t+1,b}^k$
15: $\quad$ **end for**
16: **end for**
**output:** $\bar{x}_t = \frac{1}{K} \sum_{k \in \mathcal{K}} x_t^k$

---

The updates for $\nabla_y f$ and $\nabla_{xy}^2 g$ are conducted in a similar manner (Steps 7 & 8), but it requires extra efforts to evaluate $[\nabla_{yy}^2 g]^{-1}$, because it has no unbiased estimator. To be specific, we note that $\frac{1}{K} \sum_{k \in \mathcal{K}} [\nabla_{yy}^2 g^k]^{-1} \neq [\frac{1}{K} \sum_{k \in \mathcal{K}} \nabla_{yy}^2 g^k]^{-1}$, making the unbiased estimator of the desired term unavailable even if each agent has an unbiased estimator for $[\nabla_{yy}^2 g^k]^{-1}$. This is indeed a unique challenge for decentralized bilevel optimization. To overcome this issue, we propose a *novel* approach that each agent $k$ constructs $b$ independent estimators $\{v_{t,j}^k\}_{j=1}^b$ for $\nabla_{yy}^2 g(x_t^k, y_t^k)$ using consensus and stochastic approximation. We then estimate $\nabla_{yy}^2 g(x_t^k, y_t^k)$ by utilizing the following approximation.

$$\left( \frac{1}{L_g} \nabla_{yy}^2 g(x_t^k, y_t^k) \right)^{-1} \approx \mathbf{I} + \sum_{j=1}^b \left( \mathbf{I} - \frac{1}{L_g} \nabla_{yy}^2 g(x_t^k, y_t^k) \right)^j \approx \mathbf{I} + \sum_{i=1}^b \prod_{j=1}^i \left( \mathbf{I} - \frac{1}{L_g} v_{t,j}^k \right).$$

We provide details in Steps 9 - 14.

Finally, each agent will compute the gradient (3) using the estimators obtained in the above procedure and update the outer solution $x_t^k$ by using the combination of gossip communication and stochastic gradient descent (Step 4). The inner loop solution $y_t^k$ can be updated similarly by Step 5.

We highlight the following key features of our DSBO algorithm: (1) each agent only communicates its estimates instead of the raw data in the gossip-communication process, preserving data privacy. (2) agent $k$ makes $\mathcal{O}(|\mathcal{N}_k|)$ communications with its neighbors in each round, which is much less than the total number of agents in a naive approach. (3) the algorithm is robust to contingencies in the network. If a communication channel fails, the agents can still jointly learn provided that the network is still connected. By contrast, a single-center-multi-user network would fail completely in case of a center failure.

## 5 Theory

In this section, we analyze the performance of our DSBO algorithm for both nonconvex and PL objectives and derive the convergence rates for both cases.

## 5.1 Nonconvex Objectives

We first consider the scenario where overall the objective function $F(x)$ is nonconvex. For nonconvex objectives, given the total number of iterations $T$, we employ the step-sizes in a constant form such that

$$\alpha_t = C_0 \sqrt{\tfrac{K}{T}}, \beta_t = \gamma_t = \sqrt{\tfrac{K}{T}}, \text{ and } b = \Theta(\log(T)) \text{ for all } t = 0, 1, \cdots, T, \tag{5}$$

where $C_0 > 0$ is a small constant.

**Compounded effect of consensus and SBO:** As discussed earlier, to derive the convergence rate of SBO under decentralized federated setting, the key step is to quantify the compounded effect between the consensus errors induced by the network structure and the biases induced by estimating gradient within (3). Unlike the central-server or non-federated regimes, the consensus errors induced by the decentralized network structure must be handled carefully. We conduct a thorough analysis to derive the contraction of consensus errors, and further show that both the bias and variance of the averaged estimator diminish to zero, establishing a nontrivial convergence argument for the desired gradient and Hessian. In particular, the estimators preserve a concentration property so that their variances decrease proportionally to $1/K$, suggesting that the network consensus effect does not degrade the concentration of the generated stochastic samples. To achieve the best possible convergence rate, we carefully set the algorithm parameters, including the step-sizes $\alpha_t, \gamma_t$ and averaging weights $\beta_t$, to control the above consensus errors and biases.

We provide the convergence rate of Alg. 1 in the following theorem and defer the detailed proof to Section B.7 of the supplementary material.

**Theorem 5.1** *Suppose Assumptions 3.1, 3.2, 3.3, and 3.4 hold. Letting $\bar{x}_t = \frac{1}{K} \sum_{k \in \mathcal{K}} x_t^k$, then*

$$\frac{1}{T} \sum_{t=0}^{T-1} \mathbb{E}[\|\nabla F(\bar{x}_t)\|^2] \leq \mathcal{O}\left(\frac{1}{\sqrt{KT}}\right) + \mathcal{O}\left(\frac{K}{T(1-\rho)^2}\right).$$

More details and proof are deferred to Section B.7 of the supplement.

**Effect of consensus:** In this result, the $\mathcal{O}(\frac{K}{T(1-\rho)^2})$ term represents the errors induced by the consensus of the network. Despite the depending on the network structure, this term diminishes to zero in the order of $\mathcal{O}(1/T)$, becoming a small order term when $T$ is large. Consequently, given the network topology, the asymptotic convergence behavior of DSBO is independent of the network structure, answering question (ii) raised in Section 1.

**Linear speedup:** Because each agent queries $\mathcal{O}(b)$ stochastic samples per round, clearly the required iteration and per-agent sample complexities for finding an $\epsilon$-stationary point such that $\frac{1}{T} \sum_{t=0}^{T-1} \mathbb{E}[\|\nabla F(\bar{x}_t)\|^2] \leq \epsilon$ are $\mathcal{O}(\frac{1}{K\epsilon^2})$ and $\widetilde{\mathcal{O}}(\frac{1}{K\epsilon^2})$, respectively. This result implies that our algorithm achieves a linear speed-up effect proportionate to the number of agents $K$. In other words, in the presence of more agents, each agent needs to obtain fewer stochastic samples to achieve a specified accuracy. Meanwhile, our rate also matches the best-known $\mathcal{O}(1/K\epsilon^2)$ iteration and per-agent sample complexities under the decentralized vanilla stochastic gradient descent settings (Lian et al., 2018). This is the first time such a result has been established for DSBO problems.

**Single-center-multi-user-federated SBO**: We point out that a simplified version of our algorithm solves single-center-multi-user-federated SBO, where the central server collects information directly from each agent and calculates the gradient by employing the weighted-average stochastic approximation scheme for the collected information. In such a scenario, the agents no longer communicate by gossip with neighbors but synchronously receive a common solution from the central server, so that the consensus effect disappears. In Theorem D.1, we show that a variant of our algorithm can indeed achieve the same $\mathcal{O}(\frac{1}{\sqrt{KT}})$ convergence rate in this setting.

## 5.2 PL Objectives

Next we study the case where the objective function satisfies the following $\mu$-PL condition.

**Assumption 5.2** *There exists a constant $\mu > 0$ such that the objective satisfies the PL condition:*

$$2\mu(F(x) - F^*) \leq \|\nabla F(x)\|^2.$$

Note that the class of strongly convex functions is a special case of PL functions. To utilize the $\mu$-PL property and achieve fast convergence, unlike the nonconvex case (5) where the step-sizes are set as constants depending on the total number of iterations $T$, we employ step-sizes in a diminishing form that

$$\alpha_t = \frac{2}{\mu(C_1 + t)}, \beta_t = \gamma_t = \frac{C_1}{C_1 + t}, \text{ and } b = \Theta(\log(T)) \text{ for } t \geq 1, \tag{6}$$

where $C_1 > 0$ is a large constant. By following an analytical process similar to that of the nonconvex scenario, in the next result, we derive the convergence rate of Alg. 1 for $\mu$-PL objectives.

**Theorem 5.3** *Suppose Assumptions 3.1, 3.2, 3.3, and 3.4 hold and the function satisfies the $\mu$-PL Assumption 5.2. Letting $\bar{x}_T = \frac{1}{K} \sum_{k \in \mathcal{K}} x_T^k$, then*

$$\mathbb{E}[F(\bar{x}_T)] - F^* \leq \mathcal{O}\left(\frac{1}{KT}\right) + \mathcal{O}\left(\frac{\ln T}{T^2(1-\rho)^2}\right).$$

*The iteration and per-agent sample complexities for finding an $\epsilon$-optimal point $\mathbb{E}[F(\bar{x}_T)] - F^* \leq \epsilon$ are $\mathcal{O}(\frac{1}{K\epsilon})$ and $\widetilde{\mathcal{O}}(\frac{1}{K\epsilon})$, respectively.*

Details and proof are deferred to Section C.2 of the supplement. This result shows that our algorithm achieves a faster convergence rate for functions satisfying the $\mu$-PL condition in terms of both iteration and sample complexities. First, as in the nonconvex scenario, the consensus error decays in the order of $\widetilde{\mathcal{O}}(\frac{1}{T^2(1-\rho)^2})$. Dominated by $\mathcal{O}(\frac{1}{KT})$, such on order of consensus decaying order indicates that the network structure will not affect Alg. 1's asymptotic convergence behavior under $\mu$-PL objectives. Meanwhile, the above result implies that Alg. 1 speeds up linearly with the number of agents and matches the optimal $\mathcal{O}(1/\epsilon)$ sample complexity for single-server vanilla strongly-convex stochastic optimization (Rakhlin et al., 2011). As a result, our algorithm achieves the *optimal* sample complexities for decentralized stochastic bilevel optimization, establishing the benchmark.

## 6 Numerical Experiments

In this section, we validate the practical performance of our algorithm in two examples: hyper-parameter optimization and policy evaluation in Markov Decision Processes (MDP), on artificially constructed decentralized ring networks. We run the experiments on a single server desktop computer. We provide the details of federate hyper-parameter optimization and federated policy evaluation.

**Hyper-parameter Optimization** We consider federated hyper-parameter optimization (2) for a handwriting recognition problem over the Australia handwriting dataset (Chang and Lin, 2011) consisting of data points $(w_i, z_i)$, where $w_i \in \mathbb{R}^{14}$ is the feature and $z_i \in \{0, 1\}$ indicates whether this data point belongs to category "1" or not. In our experiment, we consider the sigmoid loss function that $l_i(z) = 1/(1 + \exp(-z))$ and a strongly convex regularizer $\mathcal{R}(x, y) = \sum_{i=1}^d \frac{x_i}{2} \|y_i\|^2$. We consider a ring network of $K$ agents where each agent $i$ preserves two neighbors $(i-1)$ and $(i+1)$ and conducts a gossip communication strategy with adjacency matrix $w_{i,j} = \frac{1}{3}$ for $j \in \{i-1, i, i+1\}$.

Before testing Alg. 1, we first randomly split the dataset for training and validation, and then allocates both training and validation dataset over $K$ agents. We then run Algorithm 1 for $T = 20000$ iterations, with $b = 200$, $\alpha_t = 0.1\sqrt{K/T}$, and $\beta_t = \gamma_t = 10\sqrt{K/T}$.

To provide a benchmark for comparison, we implement a baseline algorithm Decentralized Bilevel Stochastic Approximation (DBSA) algorithm, a naive extension of the double-loop BSA algorithm (Ghadimi and Wang, 2018) in the decentralized setting, formally stated in Section E.1 of the supplementary materials.

We first consider $K = 5$, test Alg. 1 for $5 \times 10^4$ iterations, and compare its performance with DBSA. We report the validation loss against total samples in Figure 2 and observe that DSBO exhibits better performance than DBSA. In particular, Further, we observe that our algorithm outperforms the baseline algorithm DBSA in that it requires fewer samples for DSBO to achieve a certain accuracy.

To investigate the efficiency of Alg. 1 to the network structure, we test Alg. 1 over $K = 5, 10, 20$, and report the details of training and validation loss in Figure 2. Further, comparing the performances of Alg. 1 over different agents $K = 5, 10, 20$, we observe that Alg. 1 converges faster when using more agents. This observation suggests that Alg. 1 exhibits a speed-up effect when using more agents.

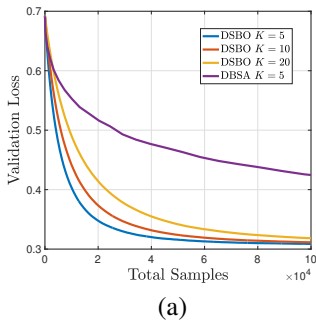
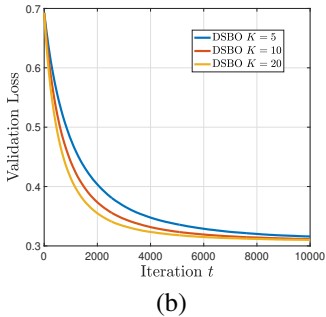

| (a) | (b) |

Figure 2: (a) Empirical averaged training loss against total samples for DSBO $K = 5, 10, 20$ and DBSA $K = 5$. (b) Empirical averaged validation loss against iteration for DSBO $K = 5, 10, 20$. All figures are generated through 10 independent simulations over the Australia handwriting dataset.

We provide additional experiments on networks of larger size ($K = 100$) and various topologies (fully-connected and randomly-connected) in Section E.1 of the supplement.

**Distributed Policy Evaluation for Reinforcement Learning** We consider a multi-agent MDP problem that arises in reinforcement learning. Let $\mathcal{S}$ be the state space. For any state $s \in \mathcal{S}$, we denote the value function by $V(s)$. We consider the scenario where the value function can be approximated by a linear function such that $V(s) = \phi_s^\top x^*$, where $\phi_s \in \mathbb{R}^m$ is a feature and $x^* \in \mathbb{R}^m$ is an unknown parameter. To obtain the optimal $x^*$, we consider the following regularized Bellman minimization problem

$$\min_x \ F(x) = \frac{1}{2|\mathcal{S}|} \sum_{s \in \mathcal{S}} \left( \phi_s^\top x - \mathbb{E}_{s'}[r(s, s') + \gamma \phi_{s'}^\top x \mid s] \right)^2 + \frac{\lambda \|x\|^2}{2},$$

where $r(s, s')$ is the random reward incurred by a transition $s$ to $s'$, $\gamma \in (0, 1)$ is the discount factor, $\lambda$ is the coefficient for the $l_2$-regularizer, and the expectation is taken over all random transitions from $s$ to $s'$.

In the federated learning setting, we consider a ring network of $K$ agents. Here each agent $k$ has access to its own data with a heterogeneous random reward function $r^k$ and can only communicate with its two neighbors $k + 1$ and $k - 1$. We denote by $y_s^*(x) = \phi_s^\top x - \mathbb{E}_{s'}[\frac{1}{K} \sum_{k \in \mathcal{K}} r^k(s, s') + \gamma \phi_{s'}^\top x \mid s]$ where $r^k(s, s')$ is the random reward function for agent $k$. The above problem can then be recast as a bilevel optimization problem

$$\min_{x \in \mathbb{R}^d} f(x, y^\star(x)) = \frac{1}{2|\mathcal{S}|} \sum_{s \in \mathcal{S}} (y_s^*(x))^2 + \frac{\lambda \|x\|^2}{2}.$$

As pointed out by (Wang et al., 2016), the above problem is $\lambda$-strongly convex.

In our experiments, we simulate an environment with state space $|\mathcal{S}| = 100$ and set the regularizer parameter $\lambda = 1$. We test the performance of Alg.1 over three cases with $K = 5, 10, 20$ and conduct 10 independent simulations for each $K$. We implement a baseline double-loop algorithm DSGD that first estimates $y_s^*(x_t)$ with $t$ samples in iteration $t$ and then optimizes the solution $x_t$. We defer the implementation details of the environment and above algorithms to Section E.2 of the supplement.

We first consider $K = 5$, run Alg. 1 for $10^4$ iterations and compare its performance with DSGD. We plot the empirical averaged mean square error $\|\bar{x}_t - x^*\|^2$ against total samples generated by *all* agents in Figure 2. This empirical result suggests that Alg. 1 outperforms the DSGD. To investigate the convergence rate of DSBO, we compare the performance of DSBO over all three scenarios with $K = 5, 10, 20$ and plot the trajectory of the averaged log-error $\log(\|\bar{x}_t - x^*\|^2)$ averaged, with one a straight line of slope -1 provided for comparison. We observe that for all four cases, the slopes of $\log(\|\bar{x}_t - x^*\|^2)$ are close to -1, matching our theoretical claim in Theorem 5.3 that Alg. 1 converges at a rate of $\mathcal{O}(1/t)$ for strongly convex objectives.

In the above experiment, we also note that Alg. 1 converges faster when using more agents. To further demonstrate the linear speedup effect, we compute the total generated samples for finding an $\epsilon$-optimal solution $\|\bar{x}_t - x^*\|^2 \leq \epsilon$ and plot the 75% confidence region of the log-sample against the number of agents $K = 5, 10, 20$ in Figure 3. We observe that it takes a roughly same number of samples to

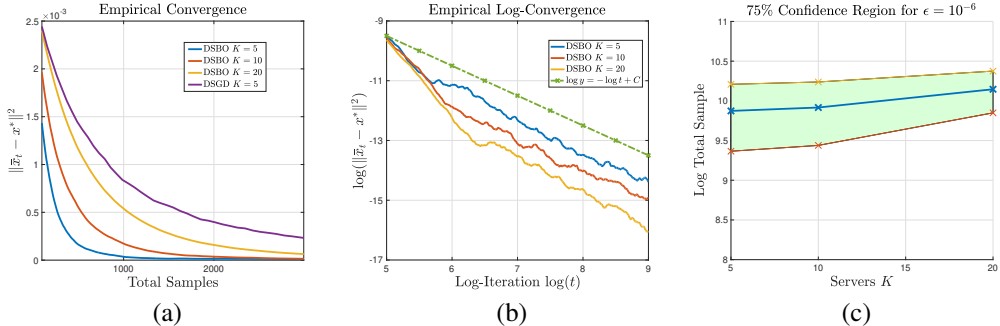

Figure 3: (a) Empirical averaged MSE$\|\bar{x}_t - x^*\|^2$ against total samples for DSBO $K = 5, 10, 20$ and DSGD $K = 5$. (b) Empirical averaged log-MSE $\log(\|\bar{x}_t - x^*\|^2)$ against log-iteration $\log(t)$ for DSBO $K = 5, 10, 20$. (c) 75% confidence region of log- total samples for achieving $\|\bar{x}_t - x^*\|^2 \leq \epsilon$, with varying network sizes $K = 5, 10, 20$. All figures are generated through 10 independent simulations.

find a $10^{-6}$-optimal solution despite different number of agents being involved. This suggests that the per-node sample complexity decreases linearly with $K$, validating the linear speedup claim in Theorem 5.3. We provide additional numerical results for other optimality level $\epsilon$ in Section E.2 of the supplementary material to further demonstrate the linear speedup effect.

## 7    Conclusion

In this paper, we propose a novel formulation for decentralized stochastic bilevel optimization. We develop a gossip-based stochastic approximation scheme to solve this problem in various settings. We show that our proposed algorithm finds a stationary point at a rate of $\mathcal{O}(\frac{1}{\sqrt{KT}})$ for nonconvex objectives, and converges to the optimal solution at a rate of $\mathcal{O}(\frac{1}{KT})$ for PL objectives. Numerical experiments on hyper-parameter optimization and multi-agent federated MDP demonstrate the practical efficiency of our algorithm, exhibit the effect of speed-up in a decentralized setting, and validate our theoretical claims. In our future work, we wish to develop algorithms that achieve lower iteration complexities and enjoy lower communication costs.

## Acknowledgments and Disclosure of Funding

Mengdi Wang acknowledges support by NSF grants DMS-1953686, IIS-2107304, CMMI-1653435, and ONR grant 1006977.

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
