## Supplementary Material

## Outline

## A Notation, Detailed Assumptions, and Technical Lemmas

For notational convenience, we denote by $\bar{u}_t = \frac{1}{K} \sum_{k=1}^{K} u_t^k$ the averaged estimates of $\nabla_{xy}^2 g(\bar{x}_t, y_t^\star)$ over $u_t^k$'s within the network. We denote by $\bar{v}_{t,j} = \frac{1}{K} \sum_{k=1}^{K} v_{t,j}^k \in \mathbb{R}^{d_y \times d_y}$, $\bar{s}_t = \frac{1}{K} \sum_{k=1}^{K} s_t^k$, and $\bar{h}_t = \frac{1}{K} \sum_{k=1}^{K} h_t^k$. We denote by $\bar{z}_k = \sum_{k=1}^{K} z_t^k$. We denote by $\|a\| = \|a\|_2$ for a vector $a$ and denote by $\|A\| = \|A\|_2$ for a matrix $A$. We denote by $\|A\|_F$ the Frobenius norm for a matrix $A$ and denote by $\langle A, B \rangle_F = \sum_{i,j} A_{ij} B_{ij}$ the Frobenius inner product for two matrices $A$ and $B$. For any $\bar{x}_t \in \mathbb{R}^{d_x}$, we denote by $y_t^\star = y^\star(\bar{x}_t)$. For notational convenience, we drop the sub-scripts $\xi^k, \zeta^k$ within the expectations $\mathbb{E}_{\xi^k}[\cdot]$ and $\mathbb{E}_{\zeta^k}[\cdot]$.

### A.1 Notation and Detailed Assumptions

In this paper, we denote by $L_q = \frac{1}{\kappa_g L_g} \geq \mu_g^{-1}$ and observe that $\|[\nabla_{yy}^2 g(x,y)]^{-1}\|_2^2 \leq \mu_g^{-2} \leq L_q^2$ for all $x \in \mathbb{R}^{d_x}$ and $y \in \mathbb{R}^{d_y}$. For notational convenience, we use $\sigma_g, \sigma_f > 0$ to represent the upper bounds of standard deviations such that

$$\mathbb{E}[\|\nabla_x f^k(x,y;\zeta^k) - \nabla_x f^k(x,y)\|^2] \leq \sigma_f^2, \mathbb{E}[\|\nabla_y f^k(x,y;\zeta^k) - \nabla_y f^k(x,y)\|^2] \leq \sigma_f^2,$$

and

$$\mathbb{E}[\|\nabla_y g^k(x,y;\xi^k) - \nabla_y g^k(x,y)\|^2] \leq \sigma_g^2,$$
$$\mathbb{E}[\|\nabla_{xy}^2 g^k(x,y;\xi^k) - \nabla_{xy}^2 g^k(x,y)\|_F^2] \leq \sigma_g^2, \mathbb{E}[\|\nabla_{yy}^2 g^k(x,y;\xi^k) - \nabla_{yy}^2 g^k(x,y)\|_F^2] \leq \sigma_g^2.$$

We also adopt constants $L_F, L_y > 0$ to quantify the Lipschitz properties, specified in Section A.2. Given $(x,y)$, we use $\nabla_x f^k(x,y;\zeta^k)$, $\nabla_y f^k(x,y;\zeta^k)$, $\nabla_y g^k(x,y;\xi^k)$, $\nabla_{xy}^2 g^k(x,y;\xi^k)$, and $\nabla_{yy}^2 g^k(x,y;\xi^k)$ to represent the independent stochastic information sampled in round $t$ by agent $k$. Such independent samples can be obtained by querying the $\mathcal{SO}$ at $(x,y)$ for three times.

### A.2 Technical Lemmas for Lipschitz Properties and Hessian Inverse Estimation

We first restate Lemmas 2.2 of (Ghadimi and Wang, 2018) to characterize the smoothness properties of $y^\star(x)$ and $\nabla F(x)$.

**Lemma A.1** *Suppose Assumptions 3.3 and 3.4 hold. Then there exist constants $L_F, L_y > 0$ dependent on the constants within Assumptions 3.3 and 3.4 such that for any $x_1, x_2 \in \mathbb{R}^{d_x}$, the followings hold.*

$$\|\nabla F(x_1) - \nabla F(x_2)\| \leq L_F \|x_1 - x_2\| \text{ and } \|y^\star(x_1) - y^\star(x_2)\| \leq L_y \|x_1 - x_2\|.$$

In this supplementary material, we adopt $L_F$ and $L_y$ to quantity the Lipschitz properties of the above functions.

**Lemma A.2** *Let $A$ be a positive definite matrix such that $\delta I \succeq A \succ 0$ for some $0 < \delta < 1$, and $A_1, \cdots, A_k$ be $k$ matrices such that $\mathbb{E}[\|\prod_{i=j}^{k} A_i\|_2^2] \leq \delta^{2(k-j+1)}$ for $1 \leq j \leq k$. Let $Q_k = I + A_k + A_{k-1}A_k + \cdots + A_1 A_2 \cdots A_k$, then the following holds.*

$$\mathbb{E}[\|(I - A)^{-1} - Q_k\|_2^2] \leq \frac{1}{1 - \delta}\left(\sum_{1 \leq j \leq k+1} \delta^{k-j}\mathbb{E}[\|A_j - A\|_2^2]\right) + \frac{\delta^{k+1}}{(1 - \delta)^3}.$$

*Proof:* Recall that for any positive definite matrix $A$ such that $\delta I \succ A \succeq 0$ for some $0 < \delta < 1$, we have

$$(I - A)^{-1} = \sum_{i=0}^{\infty} A^i = I + A + A^2 + \cdots.$$

Letting $Q = (I - A)^{-1}$, we have

$$Q = I + AQ \quad \text{and} \quad \|Q\|_2^2 = \|(I - A)^{-1}\|_2^2 \leq \frac{1}{(1 - \delta)^2}.$$

We define an auxiliary sequence $Q_1 = I + A_1$, $Q_2 = I + A_2 Q_1, \cdots$, and $Q_{k-1} = I + A_{k-1}Q_{k-2}$, and note that $Q_k = I + A_k Q_{k-1}$. Consider $(I - A)^{-1} - Q_k$, by utilizing the above sequence, we obtain

$$(I - A)^{-1} - Q_k = AQ - A_k Q_{k-1} = (A - A_k)Q + A_k(Q - Q_{k-1}) + A^{k+1}Q.$$

We note that $Q - Q_{k-1} = AQ - A_{k-1}Q_{k-2} = (A - A_{k-1})Q + A_{k-1}(Q - Q_{k-2})$. By using such induction relationship, we can quantify the estimation error by

$$\|(I - A)^{-1} - Q_k\|_2 \leq \|A - A_k\|_2\|Q\|_2 + \|A_k\|_2\|A - A_{k-1}\|_2\|Q\|_2 + \cdots$$
$$+ \|A_k A_{k-1} \cdots A_2\|_2\|A - A_1\|_2\|Q\|_2 + \|A^{k+1}Q\|_2.$$

Letting $a_i = \|A - A_i\|_2$ and $b_i = \|A_{i+1} \cdots A_k\|_2\|Q\|_2$, and taking expectations on both sides of the above inequality, we obtain $\mathbb{E}[\|b_i\|_2^2] \leq \delta^{2(k-i)}\|Q\|_2$ and $\mathbb{E}[\|A^{k+1}\|_2^2] \leq \delta^{2(k+1)}$. By using the fact that $\|AB\|_2 \leq \|A\|_2\|B\|_2 \leq \frac{\|A\|_2^2}{2} + \frac{\|B\|_2^2}{2}$, we further have that

$$\mathbb{E}[\|(I - A)^{-1} - Z_k\|_2^2]$$

$$\leq \sum_{i=1}^{k} \mathbb{E}[\|b_i a_i\|_2^2] + \sum_{1 \leq i < j \leq k} 2\mathbb{E}[\|b_i b_j a_i a_j\|_2] + \sum_{1 \leq i \leq k} 2\mathbb{E}[\|a_i b_i A^{k+1}Q\|_2] + \mathbb{E}[\|A^{k+1}Q\|_2^2]$$

$$\leq \sum_{i=1}^{k} \mathbb{E}[\|b_i\|_2^2]\mathbb{E}[\|a_i\|_2^2] + \sum_{1 \leq i < j \leq k} \mathbb{E}[\|b_i b_j\|_2]\mathbb{E}[\|a_i\|_2^2 + \|a_j\|^2]$$

$$+ \sum_{1 \leq i \leq k} 2\delta^{2k+1-i}\mathbb{E}[\|Q\|_2\|a_i\|_2] + \delta^{2(k+1)}\|Q\|_2^2$$

$$\leq \sum_{i=1}^{k} \delta^{2(k-i)}\mathbb{E}[\|a_i\|_2^2] + \sum_{1 \leq i < j \leq k} \delta^{2k-i-j}\mathbb{E}[\|a_i\|_2^2 + \|a_j\|_2^2]$$

$$+ \sum_{1 \leq i \leq k} \delta^{2k+1-i}\mathbb{E}[\|Q\|_2^2 + \|a_i\|_2^2] + \delta^{2(k+1)}\|Q\|_2^2$$

$$= \left(\sum_{1 \leq i \leq k} \delta^{k-i} + \delta^{k+1}\right)\left(\sum_{1 \leq j \leq k} \delta^{k-j}\mathbb{E}[\|a_j\|_2^2]\right) + \left(\delta^{2(k+1)} + \sum_{1 \leq i \leq k} \delta^{2k+1-i}\right)\|Q\|_2^2$$

$$\leq \frac{1}{1 - \delta}\left(\sum_{1 \leq j \leq k} \delta^{k-j}\mathbb{E}[\|a_j\|_2^2]\right) + \frac{\delta^{k+1}}{1 - \delta}\|Q\|_2^2,$$

where the last inequality uses the fact that $\sum_{i=0}^{\infty} \delta^i = \frac{1}{1-\delta}$. The desired inequality can be acquired by using $\|Q\|_2^2 \leq \frac{1}{(1-\delta)^2}$.

$\square$

We provide the following result to characterize the estimation error $\|[\nabla_{yy}^2 g(\bar{x}_t, \bar{y}_t)]^{-1} - q_t^k\|_2^2$ induced by Algorithm 1.

**Lemma A.3** *Suppose Assumptions 3.1, 3.2, 3.3, and 3.4 hold, then we have*

$$\mathbb{E}[\|[\nabla_{yy}^2 g(\bar{x}_t, y_t^\star)]^{-1} - q_t^k\|_2^2] \leq \frac{1}{L_g^4 \kappa_g}\Big( \sum_{1 \leq j \leq b} (1-\kappa_g)^{b-j} \mathbb{E}[\|\nabla_{yy}^2 g(\bar{x}_t, y_t^\star) - v_{t,i}^k\|_F^2]\Big) + \frac{(1-\kappa_g)^{b+1}}{L_g^2 \kappa_g^3}.$$

*Proof:* Recall that each $v_{t,i}^k$ is the convex combination of $\mu_g I$ and sampled hessian $\nabla_{yy}^2 g^k(x, y; \xi^k)$, under Assumption 3.4 (v) that $\mathbb{E}[\|I - \frac{1}{L_g}\nabla_{yy}^2 g^k(x, y; \xi^k)\|_2^2] \leq (1-\kappa_g)^2$, we have $\mathbb{E}[\|I - \frac{1}{L_g}v_{t,i}^k\|_2^2] \leq (1 - \kappa_g)^2$. By applying Lemma A.2 with $A = I - \frac{1}{L_g}\nabla_{yy}^2 g(\bar{x}_t, y_t^\star)$, $A_i = I - \frac{v_{t,i}^k}{L_g}$, and $\delta = 1 - \kappa_g$, we obtain that

$$\mathbb{E}[\|L_g[\nabla_{yy}^2 g(\bar{x}_t, y_t^\star)]^{-1} - Q_{t,b}^k\|_2^2]$$
$$\leq \frac{1}{\kappa_g}\Big( \sum_{1 \leq j \leq b} \frac{(1 - \kappa_g)^{b-j}}{L_g^2} \mathbb{E}[\|\nabla_{yy}^2 g(\bar{x}_t, y_t^\star) - v_{t,j}^k\|_2^2]\Big) + \frac{(1 - \kappa_g)^{b+1}}{\kappa_g^3}$$
$$\leq \frac{1}{\kappa_g}\Big( \sum_{1 \leq j \leq b} \frac{(1 - \kappa_g)^{b-j}}{L_g^2} \mathbb{E}[\|\nabla_{yy}^2 g(\bar{x}_t, y_t^\star) - v_{t,j}^k\|_F^2]\Big) + \frac{(1 - \kappa_g)^{b+1}}{\kappa_g^3}.$$

We obtain the desired result through dividing both sides of the above inequality by $L_g^2$ and using the fact that $q_t^k = Q_{t,b}^k/L_g$.

$\square$

# B    Proof of Results for Nonconvex Objectives

Throughout this section, we assume Assumptions 3.1, 3.2, 3.3, and 3.4 hold and the step-sizes follow (5) that

$$\alpha_t = C_0\sqrt{\tfrac{K}{T}}, \ \beta_t = \gamma_t = \sqrt{\tfrac{K}{T}}, \ \text{and } b = 3\lceil \log_{\frac{1}{1-\kappa_g}} T \rceil, \ \text{for all } t = 0, 1, \cdots, T,$$

where $C_0 > 0$ is a small constant and the number of iterations $T$ is large such that $\beta_t, \gamma_t \leq 1$ and

$$\Upsilon(C_0, T) = 1 - \frac{C_0 L_F}{\sqrt{T}} - 4C_0^2(L_f^2(m_1 + m_2) + \widetilde{L}_g^2(m_3 + m_4))(1 + L_y^2) - \frac{3m_5 L_y^2 C_0^2}{\mu_g^2} \geq 0,$$

with

$$m_1 = 4, \ m_2 = 12L_g^2 L_q^2, \ m_3 = 12C_f^2 L_q^2,$$
$$m_4 = 12C_f^2 L_g^{-2}\kappa_g^{-2}, \ \text{and } m_5 = 6\big(L_f^2(m_1 + m_2) + \widetilde{L}_g^2(m_3 + m_4)\big). \tag{7}$$

## B.1    Lemma B.1 and Its Proof

**Lemma B.1** *Suppose Assumptions 3.1, 3.2, 3.3, and 3.4 hold, then we have the followings.*

$$\mathbb{E}[\|s_t^k\|^2] \leq C_f^2, \ \mathbb{E}[\|h_t^k\|^2] \leq C_f^2, \mathbb{E}[\|u_t^k\|^2] \leq L_g^2, \mathbb{E}[\|q_t^k\|^2] \leq L_q^2,$$
$$\mathbb{E}[\|v_{t,j}^k\|_F^2] \leq L_g^2, \forall 1 \leq j \leq b, \ and \ \mathbb{E}[\|z_t^k\|^2] \leq 2C_f^2 + 2C_f^2 L_q^2 L_g^2. \tag{8}$$

*Proof:* We first observe that $s_t^k, h_t^k, u_t^k, v_{t,j}^k$ are convex combinations of past sampled stochastic information $\nabla_x f^k(x_t^k, y_t^k; \zeta_t^k)$, $\nabla_y f^k(x_t^k, y_t^k; \zeta_t^k)$, $\nabla_{xy}^2 g^k(x_t^k, y_t^k; \xi_t^k)$, $\nabla_{yy}^2 g^k(x_t^k, y_t^k; \xi_t^k)$, respectively. Therefore, under Assumption 3.3, for all $t \leq T$, for all $1 \leq j \leq b$, we have

$$\mathbb{E}[\|u_t^k\|^2] \leq L_g^2, \ \mathbb{E}[\|v_{t,j}^k\|^2] \leq L_g^2, \mathbb{E}[\|s_t^k\|^2] \leq C_f^2, \ \text{and } \mathbb{E}[\|h_t^k\|^2] \leq C_f^2.$$

Recall that $q_t^k = \frac{1}{L_g}\sum_{i=0}^b \prod_{j=1}^i (I - \frac{v_{t,j}^k}{L_g})$, we further obtain that

$$\mathbb{E}[\|q_t^k\|^2] = \frac{1}{L_g^2} \sum_{0 \leq i \leq b} \mathbb{E}\left( \prod_{j=1}^i (I - \tfrac{v_{t,j}^k}{L_g}) \cdot \sum_{0 \leq s \leq b} \prod_{j=1}^s (I - \tfrac{v_{t,j}^k}{L_g}) \right)$$
$$\leq \frac{1}{L_g^2} \sum_{0 \leq i \leq b} \frac{(1 - \kappa_g)^i}{\kappa_g} \leq \frac{1}{\kappa_g^2 L_g^2} = L_q^2. \tag{9}$$

By using the conditional independence of the sampled stochastic information, we have $\mathbb{E}[\|u_t^k q_t^k h_t^k\|^2] \leq C_f^2 L_q^2 L_g^2$, further implying that

$$\mathbb{E}[\|z_t^k\|^2] = \mathbb{E}[\|s_t^k - u_t^k q_t^k h_t^k\|^2] \leq 2C_f^2 + 2C_f^2 L_q^2 L_g^2.$$

This completes the proof. $\qquad\square$

## B.2   Lemma B.2 and Its Proof

We quantify the convergence behavior of consensus errors under the choices of step-sizes (5) and (6) as follows.

**Lemma B.2** *Suppose Assumptions 3.1, 3.2, 3.3, and 3.4 hold and the step-sizes satisfy $\beta_t \leq 1$ and one of the followings:*

*(i) $\alpha_t = \alpha_0$, $\beta_t = \beta_0$, and $\gamma_t = \gamma_0$, for $0 \leq t \leq T$.*

*(ii) $\lim_{t\to\infty}(\alpha_t + \beta_t + \gamma_t) = 0$, $\lim_{t\to\infty}\frac{\alpha_{t-1}}{\alpha_t} = 1$, $\lim_{t\to\infty}\frac{\beta_{t-1}}{\beta_t} = 1$, and $\lim_{t\to\infty}\frac{\gamma_{t-1}}{\gamma_t} = 1$.*

*Then we have for all $1 \leq j \leq b$,*

$$\sum_{k\in\mathcal{K}} \mathbb{E}[\|x_t^k - \bar{x}_t\|^2] \leq \mathcal{O}\left(\frac{K\alpha_t^2}{(1-\rho)^2}\right), \quad \sum_{k\in\mathcal{K}} \mathbb{E}[\|y_t^k - \bar{y}_t\|^2] \leq \mathcal{O}\left(\frac{K\gamma_t^2}{(1-\rho)^2}\right),$$

*and* $\sum_{k\in\mathcal{K}} \mathbb{E}\left[\|s_t^k - \bar{s}_t\|^2 + \|u_t^k - \bar{u}_t\|^2 + \|h_t^k - \bar{h}_t\|_F^2 + \|v_{t,j}^k - \bar{v}_{t,j}\|_F^2\right] \leq \mathcal{O}\left(\frac{K\beta_t^2}{(1-\rho)^2}\right).$

*Proof:* Recall the update rule that

$$X_{t+1} = X_t W - \alpha_t Z_t \text{ and } \bar{X}_{t+1} = \bar{X}_t - \alpha_t \bar{Z}_t,$$

by using the fact that $\bar{X}_t = X_t W^\infty$, we have

$$X_{t+1} - \bar{X}_{t+1} = X_t(W - W^\infty) - \alpha_t(Z_t - \bar{Z}_t).$$

Under Assumption 3.2 that $\|W - W^\infty\|_2 = \sqrt{\rho}$, we have

$$\|X_t W - \bar{X}_t\|_F = \|(X_t - \bar{X}_t)(W - W^\infty)\|_F = \|(W - W^\infty)^\top(X_t - \bar{X}_t)^\top\|_F$$
$$\leq \|W - W^\infty\|_2 \|X_t - \bar{X}_t\|_F \leq \sqrt{\rho}\|X_t - \bar{X}_t\|_F,$$

where the first equality uses the fact that $\bar{X}_t W = \bar{X}_t = X_t W^\infty$. Consequently, by using the fact that $\|A + B\|_F^2 \leq (1+\eta)\|A\|_F^2 + (1+\frac{1}{\eta})\|B\|_F^2$ for $\eta > 0$, we have

$$\|X_{t+1} - \bar{X}_{t+1}\|_F^2 \leq (1+\eta)\rho\|X_t - \bar{X}_t\|_F^2 + (1+\tfrac{1}{\eta})\alpha_t^2\|Z_t - \bar{Z}_t\|_F^2.$$

By setting $\eta = \frac{1-\rho}{2\rho}$, we obtain

$$\|X_{t+1} - \bar{X}_{t+1}\|_F^2 \leq \frac{1+\rho}{2}\|X_t - \bar{X}_t\|_F^2 + \frac{(1+\rho)\alpha_t^2}{1-\rho}\|Z_t - \bar{Z}_t\|_F^2.$$

Taking expectations on both sides of the above inequality and using Lemma B.1 that $\|Z_t - \bar{Z}_t\|_F^2 \leq 4K(C_f^2 + C_f^2 L_q^2 C_g^2)$ and $1 + \rho \leq 2$, we further have

$$\mathbb{E}[\|X_{t+1} - \bar{X}_{t+1}\|_F^2] \leq \frac{1+\rho}{2}\mathbb{E}[\|X_t - \bar{X}_t\|_F^2] + \frac{2\alpha_t^2}{1-\rho}\mathbb{E}[\|Z_t - \bar{Z}_t\|_F^2]$$
$$\leq \frac{1+\rho}{2}\mathbb{E}[\|X_t - \bar{X}_t\|_F^2] + \frac{8\alpha_t^2 K(C_f^2 + C_f^2 L_q^2 L_g^2)}{1-\rho}.$$

We then use an induction argument to prove the result. Suppose $\mathbb{E}[\|X_t - \bar{X}_t\|_F^2] \leq \hat{C}K\alpha_{t-1}^2$, then we have

$$\mathbb{E}[\|X_{t+1} - \bar{X}_{t+1}\|_F^2] \leq \frac{(1+\rho)\hat{C}K\alpha_{t-1}^2}{2} + \frac{8\alpha_t^2 K(C_f^2 + C_f^2 L_q^2 L_g^2)}{1-\rho}$$
$$= \alpha_t^2\left(\frac{(1+\rho)\hat{C}K\alpha_{t-1}^2}{2\alpha_t^2} + \frac{8K(C_f^2 + C_f^2 L_q^2 L_g^2)}{1-\rho}\right).$$

We observe that $\frac{(1+\rho)\alpha_{t-1}^2}{2\alpha_t^2} = \frac{1+\rho}{2}$ under condition (i). Under condition (ii) where $\lim_{t\to\infty} \frac{\alpha_{t-1}}{\alpha_t} = 1$, we can see that $\frac{(1+\rho)\alpha_{t-1}^2}{2\alpha_t^2} \leq \frac{3+\rho}{4}$ for $t$ sufficiently large. Combining both scenarios, we observe that $\left(\frac{(1+\rho)\hat{C}\alpha_{t-1}^2}{2\alpha_t^2} + \frac{8(C_f^2+C_f^2 L_q^2 L_g^2)}{1-\rho}\right) \leq \hat{C}$ for $\hat{C} = \frac{32(C_f^2+C_f^2 L_q^2 L_g^2)}{(1-\rho)^2}$. We then obtain that

$$\sum_{k\in\mathcal{K}} \mathbb{E}[\|x_t^k - \bar{x}_t\|^2] \leq \mathcal{O}\left(\frac{K\alpha_t^2}{(1-\rho)^2}\right).$$

The analysis for $\sum_{k\in\mathcal{K}} \mathbb{E}[\|y_t^k - \bar{y}_t\|^2]$ is similar. To quantify $\sum_{k\in\mathcal{K}} \mathbb{E}[\|s_t^k - \bar{s}_t\|^2]$, we observe that a weight $1 - \beta_t \leq 1$ is assigned to the prior value $s_t^k$, yielding that

$$\sum_{k\in\mathcal{K}} \mathbb{E}[\|s_{t+1}^k - \bar{s}_{t+1}\|^2] \leq \frac{(1+\rho)(1-\beta_t)^2}{2} \sum_{k\in\mathcal{K}} \mathbb{E}[\|s_t^k - \bar{s}_t\|^2] + \frac{4\beta_t^2 KC_f^2}{1-\rho}.$$

We acquire the desired result by following the analysis of quantifying $\sum_{k\in\mathcal{K}} \mathbb{E}[\|x_t^k - \bar{x}_t\|^2]$. $\qquad\square$

## B.3   Lemma B.3 and Its Proof

We recall Algorithm 1 Step 4 that $x_{t+1}^k = \sum_{j\in\mathcal{N}_k} w_{k,j} x_t^j - \alpha_t \left(s_t^k - u_t^k q_t^k h_t^k\right)$ and express $\bar{x}_{t+1}$ as follows.

$$\bar{x}_{t+1} = \bar{x}_t - \alpha_t \bar{z}_t, \text{ where } \bar{z}_t = \frac{1}{K} \sum_{k\in\mathcal{K}} \left(s_t^k - u_t^k q_t^k h_t^k\right). \tag{10}$$

**Lemma B.3** *Suppose Assumptions 3.1, 3.2, 3.3, and 3.4 hold. We have that*

$$\mathbb{E}[\|\nabla F(\bar{x}_t)\|^2]$$
$$\leq \frac{2}{\alpha_t}\left(\mathbb{E}[F(\bar{x}_t)] - \mathbb{E}[F(\bar{x}_{t+1})]\right) - (1-\alpha_t L_F)\mathbb{E}[\|\bar{z}_t\|^2] + 4\mathbb{E}[\|\nabla_x f(\bar{x}_t, y_t^\star) - \bar{s}_t\|^2]$$
$$\quad + 12 C_g^2 L_q^2 \mathbb{E}[\|\nabla_y f(\bar{x}_t, y_t^\star) - \bar{h}_t\|^2] + 12 C_f^2 L_q^2 \mathbb{E}[\|\nabla_{xy}^2 g(\bar{x}_t, y_t^\star) - \bar{u}_t\|_F^2] \tag{11}$$
$$\quad + \frac{12 C_f^2}{L_g^2 \kappa_g} \sum_{1\leq j \leq b} (1-\kappa_g)^{b-j} \mathbb{E}[\|\nabla_{yy}^2 g(\bar{x}_t, y_t^\star) - \bar{v}_{t,i}\|_F^2] + \mathcal{O}\left(\frac{\beta_t^2}{(1-\rho)^2}\right).$$

*Proof:* We start from the $L_F$-smoothness of $F(x)$ provided by Lemma A.1:

$$F(\bar{x}_{t+1}) - F(\bar{x}_t) \leq \langle \nabla F(\bar{x}_t), \bar{x}_{t+1} - \bar{x}_t \rangle + \frac{\alpha_t^2 L_F}{2}\|\bar{z}_t\|^2$$

$$\leq -\alpha_t \langle \nabla F(\bar{x}_t), \bar{z}_t \rangle + \frac{\alpha_t^2 L_F}{2}\|\bar{z}_t\|^2.$$

By using the fact that $-2\langle a, b\rangle = -\|a\|^2 - \|b\|^2 + \|a-b\|^2$, we further obtain

$$F(\bar{x}_{t+1}) - F(\bar{x}_t)$$
$$\leq -\frac{\alpha_t}{2}\|\nabla F(\bar{x}_t)\|^2 - \frac{\alpha_t}{2}\|\bar{z}_t\|^2 + \frac{\alpha_t}{2}\|\nabla F(\bar{x}_t) - \bar{z}_t\|^2 + \frac{\alpha_t^2 L_F}{2}\|\bar{z}_t\|^2. \tag{12}$$

Consider the term $\|\nabla F(\bar{x}_t) - \bar{z}_t\|^2$, we denote by $y_t^\star = y^\star(\bar{x}_t)$ and obtain

$$\|\nabla F(\bar{x}_t) - \bar{z}_t\|^2 = \|\nabla F(\bar{x}_t) - \frac{1}{K}\sum_{k\in\mathcal{K}} z_t^k\|^2$$

$$\leq \frac{2}{K}\sum_{k\in\mathcal{K}}\left(\|\nabla_x f(\bar{x}_t, y_t^\star) - s_t^k\|^2 + \|\nabla_{xy}^2 g(\bar{x}_t, y_t^\star)[\nabla_{yy}^2 g(\bar{x}_t, y_t^\star)]^{-1}\nabla_y f(\bar{x}_t, y_t^\star) - u_t^k[v_t^k]^{-1}h_t^k\|^2\right)$$

$$\leq \frac{2}{K}\sum_{k\in\mathcal{K}}\left(\|\nabla_x f(\bar{x}_t, y_t^\star) - s_t^k\|^2 + 3\|\nabla_y f(\bar{x}_t, \bar{y}_t)\|^2\|[\nabla_{yy}^2 g(\bar{x}_t, y_t^\star)]^{-1}\|^2\|\nabla_{xy}^2 g(\bar{x}_t, y_t^\star) - u_t^k\|^2\right)$$

$$\quad + \frac{6}{K}\sum_{k\in\mathcal{K}}\left(\|u_t^k\|^2\|\nabla_y f(\bar{x}_t, y_t^\star)\|^2\|[\nabla_{yy}^2 g(\bar{x}_t, y_t^\star)]^{-1} - q_t^k\|^2\right)$$

$$\quad + \frac{6}{K}\sum_{k\in\mathcal{K}}\left(\|u_t^k\|^2\|q_t^k\|^2\|\nabla_y f(\bar{x}_t, y_t^\star) - h_t^k\|^2\right).$$

Taking expectations on both sides of the above inequality and applying Lemma B.1, we have that

$$
\mathbb{E}[\|\nabla F(\bar{x}_t) - \bar{z}_t\|^2]
$$
$$
\leq \frac{2}{K} \sum_{k \in \mathcal{K}} \Big( \mathbb{E}[\|\nabla_x f(\bar{x}_t, y_t^\star) - s_t^k\|^2] + 3C_f^2 L_q^2 \mathbb{E}[\|\nabla_{xy}^2 g(\bar{x}_t, y_t^\star) - u_t^k\|^2] \Big)
$$
$$
+ \frac{6}{K} \sum_{k \in \mathcal{K}} \Big( L_g^2 C_f^2 \mathbb{E}[\|[\nabla_{yy}^2 g(\bar{x}_t, y_t^\star)]^{-1} - q_t^k\|^2] + L_g^2 L_q^2 \mathbb{E}[\|\nabla_y f(\bar{x}_t, y_t^\star) - h_t^k\|^2] \Big).
$$

Taking expectations on both sides of (12), combining with the above inequality, dividing both sides by $\frac{\alpha_t}{2}$, and rearranging the terms, we obtain

$$
\mathbb{E}[\|\nabla F(\bar{x}_t)\|^2]
$$
$$
\leq \frac{2}{\alpha_t} \Big( \mathbb{E}[F(\bar{x}_{t+1})] - \mathbb{E}[F(\bar{x}_t)] \Big) - (1 - \alpha_t L_F) \mathbb{E}[\|\bar{z}_t\|^2]
$$
$$
+ \frac{2}{K} \sum_{k \in \mathcal{K}} \Big( \mathbb{E}[\|\nabla_x f(\bar{x}_t, y_t^\star) - s_t^k\|^2] + 3C_f^2 L_q^2 \mathbb{E}[\|\nabla_{xy}^2 g(\bar{x}_t, y_t^\star) - u_t^k\|^2] \Big) \tag{13}
$$
$$
+ \frac{6}{K} \sum_{k \in \mathcal{K}} \Big( L_g^2 C_f^2 \mathbb{E}[\|[\nabla_{yy}^2 g(\bar{x}_t, y_t^\star)]^{-1} - q_t^k\|^2] + L_g^2 L_q^2 \mathbb{E}[\|\nabla_y f(\bar{x}_t, y_t^\star) - h_t^k\|^2] \Big).
$$

By applying Lemma A.3 with $(1 - \kappa_g)^{b+1} \leq \mathcal{O}(\frac{1}{T^3})$ for $b = 3\lceil \log_{\frac{1}{1-\kappa_g}}(T) \rceil$, we conclude

$$
\mathbb{E}[\|\nabla F(\bar{x}_t)\|^2]
$$
$$
\leq \frac{2}{\alpha_t} \Big( \mathbb{E}[F(\bar{x}_t)] - \mathbb{E}[F(\bar{x}_{t+1})] \Big) - (1 - \alpha_t L_F) \mathbb{E}[\|\bar{z}_t\|^2] + \frac{2}{K} \sum_{k \in \mathcal{K}} \mathbb{E}[\|\nabla_x f(\bar{x}_t, y_t^\star) - s_t^k\|^2]
$$
$$
+ \frac{6}{K} \sum_{k \in \mathcal{K}} \Big( L_g^2 L_q^2 \mathbb{E}[\|\nabla_y f(\bar{x}_t, y_t^\star) - h_t^k\|^2] + C_f^2 L_q^2 \mathbb{E}[\|\nabla_{xy}^2 g(\bar{x}_t, y_t^\star) - u_t^k\|^2] \Big)
$$
$$
+ \frac{6}{K} \sum_{k \in \mathcal{K}} \Big[ \frac{L_g^2 C_f^2}{L_g^4 \kappa_g} \Big( \sum_{1 \leq j \leq b} (1 - \kappa_g)^{b-j} \mathbb{E}[\|\nabla_{yy}^2 g(\bar{x}_t, y_t^\star) - v_{t,i}^k\|_F^2] \Big) + \mathcal{O}\Big( \frac{1}{T^3} \Big) \Big] \tag{14}
$$
$$
\leq \frac{2}{\alpha_t} \Big( \mathbb{E}[F(\bar{x}_t)] - \mathbb{E}[F(\bar{x}_{t+1})] \Big) - (1 - \alpha_t L_F) \mathbb{E}[\|\bar{z}_t\|^2] + 4 \mathbb{E}[\|\nabla_x f(\bar{x}_t, y_t^\star) - \bar{s}_t\|^2]
$$
$$
+ 12 L_g^2 L_q^2 \mathbb{E}[\|\nabla_y f(\bar{x}_t, y_t^\star) - \bar{h}_t\|^2] + 12 C_f^2 L_q^2 \mathbb{E}[\|\nabla_{xy}^2 g(\bar{x}_t, y_t^\star) - \bar{u}_t\|_F^2]
$$
$$
+ \frac{12 C_f^2}{L_g^2 \kappa_g} \sum_{1 \leq j \leq b} (1 - \kappa_g)^{b-j} \mathbb{E}[\|\nabla_{yy}^2 g(\bar{x}_t, y_t^\star) - \bar{v}_{t,i}\|_F^2] + \mathcal{O}\Big( \frac{\beta_t^2}{(1 - \rho)^2} \Big),
$$

where the last inequality uses the facts that $\|a + b\| \leq 2\|a\|^2 + 2\|b\|^2$ and $\|A\|_2 \leq \|A\|_F$ for any matrix $A$, and applies the convergence of consensus errors characterized by Lemma B.2 that

$$
\sum_{k \in \mathcal{K}} \mathbb{E}\Big( \|s_t^k - \bar{s}_t\|^2 + \|u_t^k - \bar{u}_t\|_F^2 + \|v_{t,j}^k - \bar{v}_{t,j}\|_F^2 + \|h_t^k - \bar{h}_t\|^2 \Big) \leq \mathcal{O}\Big( \frac{K \beta_t^2}{(1 - \rho)^2} \Big).
$$

This completes the proof. $\qquad \square$

## B.4 Lemma B.4 and Its Proof

Recall Algorithm 1 Step 5 that $y_{t+1}^k = \sum_{j \in \mathcal{N}_k} y_t^j - \gamma_t \nabla_y g^k(x_t^k, y_t^k; \xi_t^k)$, we may express $\bar{y}_{t+1}$ as

$$
\bar{y}_{t+1} = \bar{y}_t - \frac{\gamma_t}{K} \sum_{k \in \mathcal{K}} \nabla_y g^k(x_t^k, y_t^k; \xi_t^k).
$$

**Lemma B.4** *Suppose Assumptions 3.1, 3.2, 3.3, and 3.4 hold and $T$ is sufficiently large, we have*

$$
\mathbb{E}[\|\bar{y}_{t+1} - y^\star(\bar{x}_{t+1})\|^2]
$$
$$
\leq (1 - \gamma_t \mu_g) \mathbb{E}[\|\bar{y}_t - y^\star(\bar{x}_t)\|^2] + \mathcal{O}\Big( \frac{\gamma_t \alpha_t^2}{(1 - \rho)^2} \Big) + \frac{2\gamma_t^2 \sigma_g^2}{K} + \frac{3 L_y^2 \alpha_t^2}{\gamma_t \mu_g} \mathbb{E}[\|\bar{z}_t\|^2]. \tag{15}
$$

*Proof:* We first decompose the estimation error $\|\bar{y}_t - y^\star(\bar{x}_t)\|^2$ as

$$\|\bar{y}_{t+1} - y^\star(\bar{x}_{t+1})\|^2 \leq \left(1 + \frac{\gamma_t \mu_g}{2}\right) \|\bar{y}_{t+1} - y^\star(\bar{x}_t)\|^2 + \left(1 + \frac{2}{\gamma_t \mu_g}\right) \|y^\star(\bar{x}_t) - y^\star(\bar{x}_{t+1})\|^2. \quad (16)$$

Recall that $\bar{y}_{t+1} = \bar{y}_t - \frac{\gamma_t}{K} \sum_{k \in \mathcal{K}} \nabla_y g^k(x_t^k, y_t^k; \xi_t^k)$. Letting $\delta_t = \nabla_y g(\bar{x}_t, \bar{y}_t) - \frac{1}{K} \sum_{k \in \mathcal{K}} \nabla_y g^k(x_t^k, y_t^k; \xi_t^k)$, we obtain

$$\begin{aligned}
\|\bar{y}_{t+1} - y^\star(\bar{x}_t)\|^2 &= \|\bar{y}_t - \gamma_t \nabla_y g(\bar{x}_t, \bar{y}_t) - y^\star(\bar{x}_t) + \gamma_t \delta_t\|^2 \\
&= \|\bar{y}_t - \gamma_t \nabla_y g(\bar{x}_t, \bar{y}_t) - y^\star(\bar{x}_t)\|^2 + \gamma_t \langle \bar{y}_t - \gamma_t \nabla_y g(\bar{x}_t, \bar{y}_t) - y^\star(\bar{x}_t), \delta_t \rangle + \gamma_t^2 \|\delta_t\|^2.
\end{aligned} \quad (17)$$

We then provide bounds for the above terms. First, consider $\|\bar{y}_t - \gamma_t \nabla_y g(\bar{x}_t, \bar{y}_t) - y^\star(\bar{x}_t)\|^2$, by using the $\mu_g$-strong convexity of $g(\bar{x}_t, y)$ in $y$ under Assumption 3.4 (i), we have

$$\begin{aligned}
&\|\bar{y}_t - \gamma_t \nabla_y g(\bar{x}_t, \bar{y}_t) - y^\star(\bar{x}_t)\|^2 \\
&\leq \|\bar{y}_t - y^\star(\bar{x}_t)\|^2 - 2\gamma_t \langle \bar{y}_t - y^\star(\bar{x}_t), \nabla_y g(\bar{x}_t, \bar{y}_t) \rangle + \gamma_t^2 \|\nabla_y g(\bar{x}_t, \bar{y}_t)\|^2 \\
&\leq (1 - 2\gamma_t \mu_g) \|\bar{y}_t - y^\star(\bar{x}_t)\|^2 + \gamma_t^2 C_g^2.
\end{aligned} \quad (18)$$

Next, consider $\langle \bar{y}_t - \gamma_t \nabla_y g(\bar{x}_t, \bar{y}_t) - y^\star(\bar{x}_t), \delta_t \rangle$, we can see that

$$\begin{aligned}
&\mathbb{E}\left[ \langle \bar{y}_t - \gamma_t \nabla_y g(\bar{x}_t, \bar{y}_t) - y^\star(\bar{x}_t), \delta_t \rangle \right] \\
&= \mathbb{E}\left[ \left(\bar{y}_t - \gamma_t \nabla_y g(\bar{x}_t, \bar{y}_t) - y^\star(\bar{x}_t)\right)^\top \left(\nabla_y g(\bar{x}_t, \bar{y}_t) - \frac{1}{K} \sum_{k=1}^{k} \nabla_y g^k(x_t^k, y_t^k)\right) \right] \\
&\leq \frac{\mu_g}{2} \mathbb{E}[\|\bar{y}_t - \gamma_t \nabla_y g(\bar{x}_t, \bar{y}_t) - y^\star(\bar{x}_t)\|^2] + \frac{\Delta_t}{2\mu_g},
\end{aligned} \quad (19)$$

where $\Delta_t = \mathbb{E}[\|\nabla_y g(\bar{x}_t, \bar{y}_t) - \frac{1}{K} \sum_{k \in \mathcal{K}} \nabla_y g^k(x_t^k, y_t^k)\|^2]$ and the last inequality comes from the fact that $\langle a, b \rangle \leq \frac{\|a\|^2}{2} + \frac{\|b\|^2}{2}$. Further, for $\|\delta_t^k\|^2$, we have

$$\begin{aligned}
\mathbb{E}[\|\delta_t^k\|^2] &= \mathbb{E}\left[ \left\| \frac{1}{K} \sum_{k \in \mathcal{K}} \left(\nabla_y g^k(\bar{x}_t, \bar{y}_t) - \nabla_y g^k(x_t^k, y_t^k; \xi_t^k)\right) \right\|^2 \right] \\
&\leq 2\mathbb{E}\left[ \left\| \frac{1}{K} \sum_{k \in \mathcal{K}} \left(\nabla_y g^k(x_t^k, y_t^k) - \nabla_y g^k(x_t^k, y_t^k; \xi_t^k)\right) \right\|^2 \right] + 2\Delta_t^k \leq \frac{2\sigma_g^2}{K} + 2\Delta_t^k,
\end{aligned} \quad (20)$$

where the last inequality uses the fact that $\{\nabla_y g^k(x_t^k, y_t^k) - \nabla_y g^k(x_t^k, y_t^k; \xi_t^k)\}$'s are conditionally mean-zero and independent such that for $k \neq s$,

$$\mathbb{E}\left[ \langle \nabla_y g^k(\bar{x}_t, \bar{y}_t) - \nabla_y g^k(x_t^k, y_t^k; \xi_t^k), \nabla_y g^s(\bar{x}_t, \bar{y}_t) - \nabla_y g^s(x_t^s, y_t^s; \xi_t^s) \rangle \right] = 0.$$

Taking expectations on both sides of (17) and combining with (18), (19), and (20), we have

$$\begin{aligned}
&\mathbb{E}[\|\bar{y}_{t+1} - y^\star(\bar{x}_t)\|^2] \\
&\leq (1 + \frac{\gamma_t \mu_g}{2})\left((1 - 2\gamma_t \mu_g)\mathbb{E}[\|\bar{y}_t - y^\star(x_t)\|^2] + \gamma_t^2 C_g^2\right) + \left(\frac{\gamma_k}{2\mu_g} + 2\gamma_t^2\right) \Delta_t + \frac{2\gamma_t^2 \sigma_g^2}{K} \\
&\leq (1 - \frac{3\gamma_t \mu_g}{2})\mathbb{E}[\|\bar{y}_t - y^\star(x_t)\|^2] + (1 + \gamma_t \mu_g)\gamma_t^2 C_g^2 \\
&\quad + \left(\frac{\gamma_k}{2\mu_g} + 2\gamma_t^2\right) \frac{1}{K} \sum_{k \in \mathcal{K}} L_g^2 \mathbb{E}\left(\|x_t^k - \bar{x}_t\|^2 + \|y_t^k - \bar{y}_t\|^2\right) + \frac{2\gamma_t^2 \sigma_g^2}{K},
\end{aligned} \quad (21)$$

where the second inequality uses the $L_g$-smoothness of $\nabla_y g(x, y)$ in both $x$ and $y$ such that

$$\begin{aligned}
\Delta_t &= \mathbb{E}\left[ \left\| \nabla_y g(\bar{x}_t, \bar{y}_t) - \frac{1}{K} \sum_{k \in \mathcal{K}} \nabla_y g^k(x_t^k, y_t^k) \right\|^2 \right] \\
&\leq \frac{1}{K} \sum_{k \in \mathcal{K}} \mathbb{E}[\|\nabla_y g^k(\bar{x}_t, \bar{y}_t) - \nabla_y g^k(x_t^k, y_t^k)\|^2] \\
&\leq \frac{1}{K} \sum_{k \in \mathcal{K}} L_g^2 \left(\mathbb{E}[\|x_t^k - \bar{x}_t\|^2] + \mathbb{E}[\|y_t^k - \bar{y}_t\|^2]\right).
\end{aligned} \quad (22)$$

Moreover, by using the $L_y$-smoothness of $y^\star(\bullet)$ characterized by Lemma A.1, we have

$$\mathbb{E}[\|y^\star(\bar{x}_t) - y^\star(\bar{x}_{t+1})\|^2] \leq L_y^2 \mathbb{E}[\|\bar{x}_t - \bar{x}_{t+1}\|^2]. \tag{23}$$

Finally, by substituting (21) and (23) into (16) and applying the bounds of the consensus errors provided by Lemma B.2, we conclude that

$$\begin{aligned}
&\mathbb{E}[\|\bar{y}_{t+1} - y^\star(\bar{x}_{t+1})\|^2] \\
&\leq (1 - \gamma_t \mu_g) \mathbb{E}[\|\bar{y}_t - y^\star(\bar{x}_t)\|^2] + (1 + \gamma_t \mu_g / 2)(1 + \gamma_t \mu_g) \gamma_t^2 C_g^2 \\
&\quad + \left( \frac{\gamma_k}{2\mu_g} + 2\gamma_t^2 \right) \frac{1}{K} \sum_{k \in \mathcal{K}} L_g^2 \left( \mathbb{E}[\|x_t^k - \bar{x}_t\|^2] + \mathbb{E}[\|y_t^k - \bar{y}_t\|^2] \right) \\
&\quad + \frac{2\gamma_t^2 \sigma_g^2}{K} + \left( 1 + \frac{2}{\gamma_t \mu_g} \right) L_y^2 \mathbb{E}[\|\bar{x}_t - \bar{x}_{t+1}\|^2] \\
&\leq (1 - \gamma_t \mu_g) \mathbb{E}[\|\bar{y}_t - y^\star(\bar{x}_t)\|^2] + \mathcal{O}(\gamma_t^3) + \mathcal{O}\left( \frac{\gamma_t \alpha_t^2}{(1 - \rho)^2} \right) + \frac{2\gamma_t^2 \sigma_g^2}{K} + \frac{3L_y^2 \alpha_t^2}{\gamma_t \mu_g} \mathbb{E}[\|\bar{z}_t\|^2],
\end{aligned}$$

where the last inequality uses the facts that $\bar{x}_{t+1} = \bar{x}_t - \alpha_t \bar{z}_t$ and $\gamma_t \mu_g \leq 1$ for large $T$. This completes the proof. $\qquad\square$

## B.5 Lemma B.5 and Its Proof

By recalling Algorithm 1 Step 6 that $s_{t+1}^k = (1 - \beta_t) \sum_{j \in \mathcal{N}_k} w_{kj} s_t^j + \beta_t \nabla_x f^k(x_t^k, y_t^k; \zeta_t^k)$, we express the update rule of $\bar{s}_t = \frac{1}{K} \sum_{k \in \mathcal{K}} s_t^k$ as

$$\bar{s}_{t+1} = (1 - \beta_t) \bar{s}_t + \frac{\beta_t}{K} \sum_{k \in \mathcal{K}} \nabla_x f^k(x_t^k, y_t^k; \zeta_t^k). \tag{24}$$

**Lemma B.5** *Suppose Assumptions 3.1, 3.2, 3.3, and 3.4 hold and $T$ is sufficiently large, then we have*

$$\begin{aligned}
&\mathbb{E}[\|\bar{s}_{t+1} - \nabla_x f(\bar{x}_{t+1}, y^\star(\bar{x}_{t+1}))\|^2] \\
&\leq (1 - \beta_t) \mathbb{E}[\|\bar{s}_t - \nabla f(\bar{x}_t, y_t^\star)\|^2] + \frac{2\beta_t^2 \sigma_f^2}{K} + \frac{4\alpha_t^2 L_f^2 (1 + L_y^2)}{\beta_t} \mathbb{E}[\|\bar{z}_t\|^2] \\
&\quad + \mathcal{O}\left( \frac{\beta_t(\alpha_t^2 + \beta_t^2)}{(1 - \rho)^2} \right) + 6\beta_t L_f^2 \mathbb{E}[\|\bar{y}_t - y_t^\star\|^2].
\end{aligned} \tag{25}$$

*Proof:* We denote by $y_t^\star = y^\star(\bar{x}_t)$ for notational convenience. Consider the update rule of $\bar{s}_{t+1}$, we have

$$\bar{s}_{t+1} - \nabla_x f(\bar{x}_{t+1}, y_{t+1}^\star) = (1 - \beta_t)[\bar{s}_t - \nabla_x f(\bar{x}_{t+1}, y_{t+1}^\star)] + \beta_t \Delta_{f,t},$$

where

$$\Delta_{f,t} = \frac{1}{K} \sum_{k \in \mathcal{K}} \nabla_x f^k(x_t^k, y_t^k; \zeta_t^k) - \nabla_x f(\bar{x}_{t+1}, y_{t+1}^\star).$$

We can see that

$$
\begin{aligned}
&\mathbb{E}[\|\bar{s}_{t+1} - \nabla_x f(\bar{x}_{t+1}, y^\star_{t+1})\|^2] \\
&= (1-\beta_t)^2 \mathbb{E}[\|\bar{s}_t - \nabla_x f(\bar{x}_{t+1}, y^\star_{t+1})\|^2] + \beta_t^2 \mathbb{E}[\|\Delta_{f,t}\|^2] \\
&\quad + 2(1-\beta_t)\beta_t \mathbb{E}\left[ (\bar{s}_t - \nabla_x f(\bar{x}_{t+1}, y^\star_{t+1}))^\top \Delta_{f,t} \right] \\
&= (1-\beta_t)^2 \mathbb{E}[\|\bar{s}_t - \nabla_x f(\bar{x}_{t+1}, y^\star_{t+1})\|^2] + \beta_t^2 \mathbb{E}[\|\Delta_{f,t}\|^2] \\
&\quad + 2(1-\beta_t)\beta_t \mathbb{E}\left[ (\bar{s}_t - \nabla_x f(\bar{x}_{t+1}, y^\star_{t+1}))^\top \Big( \frac{1}{K} \sum_{k\in\mathcal{K}} \nabla_x f^k(x_t^k, y_t^k) - \nabla_x f(\bar{x}_{t+1}, y^\star_{t+1}) \Big) \right] \\
&\leq (1-\beta_t)^2 \mathbb{E}[\|\bar{s}_t - \nabla_x f(\bar{x}_{t+1}, y^\star_{t+1})\|^2] + \beta_t^2 \mathbb{E}[\|\Delta_{f,t}\|^2] \\
&\quad + \frac{(1-\beta_t)\beta_t}{2} \Big( \mathbb{E}[\|\bar{s}_t - \nabla_x f(\bar{x}_{t+1}, y^\star_{t+1})\|^2 + 4\mathbb{E}[\| \frac{1}{K} \sum_{k\in\mathcal{K}} \nabla_x f^k(x_t^k, y_t^k) - \nabla_x f(\bar{x}_{t+1}, y^\star_{t+1})\|^2] \Big) \\
&\leq (1-\beta_t)\left(1 - \frac{\beta_t}{2}\right) \mathbb{E}[\|\bar{s}_t - \nabla_x f(\bar{x}_{t+1}, y^\star_{t+1})\|^2] + \beta_t^2 \mathbb{E}[\|\Delta_{f,t}\|^2] \\
&\quad + \frac{2\beta_t(1-\beta_t)L_f^2}{K} \sum_{k\in\mathcal{K}} \left( \mathbb{E}[\|x_t^k - \bar{x}_{t+1}\|^2] + \mathbb{E}[\|y_t^k - y^\star_{t+1}\|^2] \right),
\end{aligned}
$$

(26)

where the first equality uses the conditional independence between $\{\nabla_x f^k(x_t^k, y_t^k; \zeta_t^k) - \nabla_x f^k(x_t^k, y_t^k)\}$ and $\bar{x}_{t+1}, \bar{y}_{t+1}, \bar{s}_t$, the first inequality uses the fact that $2\langle a, b\rangle \leq \frac{\|a\|^2}{2} + 2\|b\|^2$, and the last inequality applies similar analysis as (22) under the $L_f$-smoothness of $\nabla_x f^k$. We observe that

$$
\begin{aligned}
&\|\Delta_{f,t}\|^2 \\
&= \left\| \frac{1}{K} \sum_{k\in\mathcal{K}} \left( \nabla_x f^k(x_t^k, y_t^k; \zeta_t^k) - \nabla_x f^k(x_t^k, y_t^k) \right) + \frac{1}{K} \sum_{k\in\mathcal{K}} (\nabla_x f^k(x_t^k, y_t^k) - \nabla_x f^k(\bar{x}_{t+1}, y^\star_{t+1})) \right\|^2 \\
&\leq 2\left\| \frac{1}{K} \sum_{k\in\mathcal{K}} \left( \nabla_x f^k(x_t^k, y_t^k; \zeta_t^k) - \nabla_x f^k(x_t^k, y_t^k) \right) \right\|^2 \\
&\quad + 2\left\| \frac{1}{K} \sum_{k\in\mathcal{K}} (\nabla_x f^k(x_t^k, y_t^k) - \nabla_x f^k(\bar{x}_{t+1}, y^\star_{t+1})) \right\|^2 .
\end{aligned}
$$

By noting that $\nabla_x f^k(x_t^k, y_t^k; \zeta_t^k) - \nabla_x f^k(x_t^k, y_t^k)$ is conditionally mean-zero and

$$
\mathbb{E}\left[ \left(\nabla_x f^i(x_t^i, y_t^i; \zeta_t^i) - \nabla_x f^i(x_t^i, y_t^i)\right)^\top \left(\nabla_x f^j(x_t^j, y_t^j; \zeta_t^j) - \nabla_x f^j(x_t^i, y_t^i)\right) \right] = 0, \text{ for } 1 \leq i \neq j \leq K,
$$

and taking expectations on both sides of the above inequality, we obtain

$$
\begin{aligned}
\mathbb{E}[\|\Delta_{f,t}\|^2] &\leq \frac{2}{K^2} \sum_{k\in\mathcal{K}} \mathbb{E}[\|\nabla_x f^k(x_t^k, y_t^k; \zeta_t^k) - \nabla_x f^k(x_t^k, y_t^k)\|^2] \\
&\quad + \frac{2}{K} \sum_{k\in\mathcal{K}} \mathbb{E}[\|\nabla_x f^k(x_t^k, y_t^k) - \nabla_x f^k(\bar{x}_{t+1}, y^\star_{t+1})\|^2] \\
&\leq \frac{2\sigma_f^2}{K} + \frac{2L_f^2}{K} \sum_{k\in\mathcal{K}} \left( \mathbb{E}[\|x_t^k - \bar{x}_{t+1}\|^2] + \mathbb{E}[\|y_t^k - y^\star_{t+1}\|^2] \right),
\end{aligned}
$$

(27)

where the last inequality uses the $L_f$-smoothness of $\nabla_x f^k(x, y)$ in both $x$ and $y$, similar as (22).

Further, we have

$$\|\bar{s}_t - \nabla_x f(\bar{x}_{t+1}, y^{\star}_{t+1})\|^2$$

$$\leq \left(1 + \frac{\beta_t}{3}\right) \|\bar{s}_t - \nabla_x f(\bar{x}_t, y^{\star}_t)\|^2 + \left(1 + \frac{3}{\beta_t}\right) \|\nabla_x f(\bar{x}_t, y^{\star}_t) - \nabla_x f(\bar{x}_{t+1}, y^{\star}_{t+1})\|^2$$

$$\leq \left(1 + \frac{\beta_t}{3}\right) \|\bar{s}_t - \nabla_x f(\bar{x}_t, y^{\star}_t)\|^2 + \left(1 + \frac{3}{\beta_t}\right) L_f^2(\|\bar{x}_t - \bar{x}_{t+1}\|^2 + \|y^{\star}_t - y^{\star}_{t+1}\|^2)$$

$$= \left(1 + \frac{\beta_t}{3}\right) \|\bar{s}_t - \nabla_x f(\bar{x}_t, y^{\star}_t)\|^2 + \left(1 + \frac{3}{\beta_t}\right) \alpha_t^2 L_f^2(1 + L_y^2)\|\bar{z}_t\|^2,$$

where the first inequality uses the $L_f$-smoothness of $\nabla_x f(x, y)$ and the second inequality uses the $L_y$-Lipschitz continuity of $y^{\star}(x)$ such that $\|y^{\star}(\bar{x}_t) - y^{\star}(\bar{x}_{t+1})\| \leq L_y\|\bar{x}_t - \bar{x}_{t+1}\|$ and the fact that $\alpha_t\|\bar{z}_t\| = \|\bar{x}_t - \bar{x}_{t+1}\|$.

By substituting the above inequality and (27) into (26), we obtain

$$\mathbb{E}[\|\bar{s}_{t+1} - \nabla f_x(\bar{x}_{t+1}, y^{\star}_{t+1})\|^2]$$

$$\leq (1 - \beta_t)\mathbb{E}[\|\bar{s}_t - \nabla_x f(\bar{x}_t, y^{\star}_t)\|^2] + \frac{2\beta_t^2\sigma_f^2}{K} + \frac{7\alpha_t^2 L_f^2(1 + L_y^2)}{2\beta_t}\mathbb{E}[\|\bar{z}_t\|^2]$$

$$+ \frac{2\beta_t L_f^2}{K} \sum_{k \in \mathcal{K}} \left(\|x_t^k - \bar{x}_{t+1}\|^2 + \|y_t^k - y^{\star}_{t+1}\|^2\right),$$

where the last inequality uses the fact that $(1 - \frac{\beta_t}{2})(1 + \frac{\beta_t}{3}) \leq 1$ and $(1 - \beta_t)(1 - \frac{\beta_t}{2})(1 + \frac{3}{\beta_t}) \leq \frac{7}{2\beta_t}$ when $T$ is sufficiently large.

Further, we can see

$$\sum_{k \in \mathcal{K}} \left(\mathbb{E}[\|x_t^k - \bar{x}_{t+1}\|^2] + \mathbb{E}[\|y_t^k - y^{\star}_{t+1}\|^2]\right)$$

$$\leq \sum_{k \in \mathcal{K}} \mathbb{E}\left[2\|x_t^k - \bar{x}_t\|^2 + 2\|\bar{x}_t - \bar{x}_{t+1}\|^2 + 3\|y_t^k - \bar{y}_t\|^2 + 3\|\bar{y}_t - y^{\star}_t\|^2 + 3\|y^{\star}_t - y^{\star}_{t+1}\|^2\right]$$

$$\leq \mathcal{O}\left(\frac{K(\alpha_t^2 + \beta_t^2)}{(1 - \rho)^2}\right) + 2K(1 + L_y^2)\mathbb{E}[\|\bar{x}_t - \bar{x}_{t+1}\|^2] + 3\sum_{k \in \mathcal{K}} \mathbb{E}[\|\bar{y}_t - y^{\star}_t\|^2] \qquad (28)$$

$$\leq \mathcal{O}\left(\frac{K(\alpha_t^2 + \beta_t^2)}{(1 - \rho)^2}\right) + 2K(1 + L_y^2)\alpha_t^2\mathbb{E}[\|\bar{z}_t\|^2] + 3K\mathbb{E}[\|\bar{y}_t - y^{\star}_t\|^2],$$

implying that

$$\mathbb{E}[\|\bar{s}_{t+1} - \nabla f_x(\bar{x}_{t+1}, y^{\star}_{t+1})\|^2]$$

$$\leq (1 - \beta_t)\mathbb{E}[\|\bar{s}_t - \nabla f(\bar{x}_t, y^{\star}_t)\|^2] + \frac{2\beta_t^2\sigma_f^2}{K} + \frac{4\alpha_t^2 L_f^2(1 + L_y^2)}{\beta_t}\mathbb{E}[\|\bar{z}_t\|^2]$$

$$+ \mathcal{O}\left(\frac{\beta_t(\alpha_t^2 + \beta_t^2)}{(1 - \rho)^2}\right) + 6\beta_t L_f^2\mathbb{E}[\|\bar{y}_t - y^{\star}_t\|^2].$$

This completes the proof.

$\square$

## B.6 Lemmas B.6 and Its Proof

**Lemma B.6** *Suppose Assumptions 3.1, 3.2, 3.3, and 3.4 hold and $T$ is sufficiently large, then we have*
*(a)*

$$\mathbb{E}[\|\bar{h}_{t+1} - \nabla_y f(\bar{x}_{t+1}, y^{\star}(\bar{x}_{t+1}))\|^2]$$

$$\leq (1 - \beta_t)\mathbb{E}[\|\bar{h}_t - \nabla_y f(\bar{x}_t, y^{\star}(x_t))\|^2] + \frac{4\alpha_t^2 L_f^2(1 + L_y^2)}{\beta_t}\mathbb{E}[\|\bar{z}_t\|^2] \qquad (29)$$

$$+ \frac{2\beta_t^2\sigma_f^2}{K} + \mathcal{O}\left(\frac{\beta_t(\alpha_t^2 + \beta_t^2)}{(1 - \rho)^2}\right) + 6\beta_t L_f^2\mathbb{E}[\|\bar{y}_t - y^{\star}(x_t)\|^2].$$

*(b)*

$$\mathbb{E}[\|\bar{u}_{t+1} - \nabla_{xy}^2 g(\bar{x}_{t+1}, y^\star(\bar{x}_{t+1}))\|_F^2]$$

$$\leq (1 - \beta_t)\mathbb{E}[\|\bar{u}_t - \nabla_{xy}^2 g(\bar{x}_t, y^\star(x_t))\|_F^2] + \frac{4\alpha_t^2 \widetilde{L}_g^2(1 + L_y^2)}{\beta_t}\mathbb{E}[\|\bar{z}_t\|^2] \tag{30}$$

$$+ \frac{2\beta_t^2 \sigma_g^2}{K} + \mathcal{O}\left(\frac{\beta_t(\alpha_t^2 + \beta_t^2)}{(1 - \rho)^2}\right) + 6\beta_t \widetilde{L}_g^2 \mathbb{E}[\|\bar{y}_t - y^\star(x_t)\|^2].$$

*(c)*

$$\mathbb{E}[\|\bar{v}_{t+1} - \nabla_{yy}^2 g(\bar{x}_{t+1}, y^\star(\bar{x}_{t+1}))\|_F^2]$$

$$\leq (1 - \beta_t)\mathbb{E}[\|\bar{v}_t - \nabla_{yy}^2 g(\bar{x}_t, y^\star(x_t))\|_F^2] + \frac{4\alpha_t^2 \widetilde{L}_g^2(1 + L_y^2)}{\beta_t}\mathbb{E}[\|\bar{z}_t\|^2] \tag{31}$$

$$+ \frac{2\beta_t^2 \sigma_g^2}{K} + \mathcal{O}\left(\frac{\beta_t(\alpha_t^2 + \beta_t^2)}{(1 - \rho)^2}\right) + 6\beta_t \widetilde{L}_g^2 \mathbb{E}[\|\bar{y}_t - y^\star(x_t)\|^2].$$

*Proof:* Part (a) can be derived by following the analysis of Lemma B.5. The key difference between parts (b)-(c) and Lemma B.5 is that here we establish stochastic recursions in terms of the Frobenius norm of the estimation error matrices $\nabla_{xy}^2 g(\bar{x}_t, \bar{y}_t) - \bar{u}_t \in \mathbb{R}^{d_x \times d_y}$ and $\nabla_{yy}^2 g(\bar{x}_t, \bar{y}_t) - \bar{v}_{t,i} \in \mathbb{R}^{d_y \times d_y}$, while Lemma B.5 considers the Euclidean norm of the estimation error vector $\nabla_x f(\bar{x}_t, \bar{y}_t) - \bar{s}_t \in \mathbb{R}^{d_x}$.

(b) Here we provide the detailed proof of part (b) for completeness. By recalling Algorithm 1 Step 8 that $u_{t+1}^k = (1 - \beta_t) \sum_{j \in \mathcal{N}_k} w_{kj} u_t^j + \beta_t \nabla_{xy}^2 g^k(x_t^k, y_t^k; \xi_t^k)$, we express the update rule of $\bar{u}_t = \frac{1}{K} \sum_{k \in \mathcal{K}} u_t^k$ as

$$\bar{u}_{t+1} = (1 - \beta_t)\bar{u}_t + \frac{\beta_t}{K} \sum_{k \in \mathcal{K}} \nabla_{xy}^2 g^k(x_t^k, y_t^k; \xi_t^k).$$

Equivalently, we have

$$\bar{u}_{t+1} - \nabla_{yx}^2 g(\bar{x}_{t+1}, y_{t+1}^\star) = (1 - \beta_t)[\bar{u}_t - \nabla_{xy}^2 g(\bar{x}_{t+1}, y_{t+1}^\star)] + \beta_t \Delta_{g,t},$$

where

$$\Delta_{g,t} = \frac{1}{K} \sum_{k \in \mathcal{K}} \nabla_{yx}^2 g^k(x_t^k, y_t^k; \xi_t^k) - \nabla_{yx}^2 g(\bar{x}_{t+1}, y_{t+1}^\star).$$

Following (26), we can see that

$$\mathbb{E}[\|\bar{u}_{t+1} - \nabla_{xy}^2 g(\bar{x}_{t+1}, y_{t+1}^\star)\|_F^2]$$

$$= (1 - \beta_t)^2 \mathbb{E}[\|\bar{u}_t - \nabla_{xy}^2 g(\bar{x}_{t+1}, y_{t+1}^\star)\|_F^2] + \beta_t^2 \mathbb{E}[\|\Delta_{g,t}\|_F^2]$$

$$+ \frac{2(1 - \beta_t)\beta_t}{K}\mathbb{E}\left[\left\langle \bar{s}_t - \nabla_x f(\bar{x}_{t+1}, y_{t+1}^\star), \sum_{k \in \mathcal{K}} \nabla_x f^k(x_t^k, y_t^k) - \nabla_x f(\bar{x}_{t+1}, y_{t+1}^\star)\right\rangle_F\right]$$

$$\leq (1 - \beta_t)\left(1 - \frac{\beta_t}{2}\right)\mathbb{E}[\|\bar{u}_t - \nabla_{xy}^2 g(\bar{x}_{t+1}, y_{t+1}^\star)\|_F^2] + \beta_t^2 \mathbb{E}[\|\Delta_{g,t}\|_F^2] \tag{32}$$

$$+ 2(1 - \beta_t)\beta_t \mathbb{E}[\|\frac{1}{K} \sum_{k \in \mathcal{K}} \nabla_x f^k(x_t^k, y_t^k) - \nabla_x f(\bar{x}_{t+1}, y_{t+1}^\star)\|_F^2]$$

$$\leq (1 - \beta_t)\left(1 - \frac{\beta_t}{2}\right)\mathbb{E}[\|\bar{u}_t - \nabla_{xy}^2 g(\bar{x}_{t+1}, y_{t+1}^\star)\|_F^2] + \beta_t^2 \mathbb{E}[\|\Delta_{g,t}\|_F^2]$$

$$+ \frac{2\beta_t(1 - \beta_t)L_g^2}{K} \sum_{k \in \mathcal{K}} \left(\mathbb{E}[\|x_t^k - \bar{x}_{t+1}\|^2] + \mathbb{E}[\|y_t^k - y_{t+1}^\star\|^2]\right),$$

where $\langle \cdot, \cdot \rangle_F$ is the Frobenius inner product. By following (27), we may obtain

$$
\begin{aligned}
\mathbb{E}[\|\Delta_{g,t}\|_F^2] \leq{} & \frac{2}{K^2} \sum_{k \in \mathcal{K}} \mathbb{E}[\|\nabla^2_{xy} g^k(x_t^k, y_t^k; \xi_t^k) - \nabla^2_{xy} g^k(x_t^k, y_t^k)\|_F^2] \\
& + \frac{2}{K} \sum_{k \in \mathcal{K}} \mathbb{E}[\|\nabla^2_{xy} g^k(x_t^k, y_t^k) - \nabla^2_{xy} g^k(\bar{x}_{t+1}, y_{t+1}^\star)\|_F^2] \\
\leq{} & \frac{2\sigma_g^2}{K} + \frac{2\widetilde{L}_g^2}{K} \sum_{k \in \mathcal{K}} \left( \mathbb{E}[\|x_t^k - \bar{x}_{t+1}\|^2] + \mathbb{E}[\|y_t^k - y_{t+1}^\star\|^2] \right).
\end{aligned}
\tag{33}
$$

Using the fact that $\|A + B\|_F^2 \leq (1 + \frac{\beta_t}{3}) \|A\|_F^2 + (1 + \frac{3}{\beta_t}) \|B\|_F^2$ and Lemma A.1 that $\|y_{t+1}^\star - y_t^\star\| \leq L_y \|x_{t+1} - x_t\|$, we have

$$
\begin{aligned}
& \|\bar{u}_t - \nabla^2_{xy} g(\bar{x}_{t+1}, y_{t+1}^\star)\|^2 \\
& \leq \left( 1 + \frac{\beta_t}{3} \right) \|\bar{u}_t - \nabla^2_{xy} g(\bar{x}_t, y_t^\star)\|^2 + \left( 1 + \frac{3}{\beta_t} \right) \|\nabla^2_{xy} g(\bar{x}_t, y_t^\star) - \nabla^2_{xy} g(\bar{x}_t, y_t^\star)\|_F^2 \\
& \leq \left( 1 + \frac{\beta_t}{3} \right) \|\bar{u}_t - \nabla^2_{xy} g(\bar{x}_t, y_t^\star)\|^2 + \left( 1 + \frac{3}{\beta_t} \right) \widetilde{L}_g^2 (\|\bar{x}_t - \bar{x}_{t+1}\|^2 + \|y_t^\star - y_{t+1}^\star\|^2) \\
& = \left( 1 + \frac{\beta_t}{3} \right) \|\bar{u}_t - \nabla^2_{xy} g(\bar{x}_t, y_t^\star)\|^2 + \left( 1 + \frac{3}{\beta_t} \right) \alpha_t^2 \widetilde{L}_g^2 (1 + L_y^2) \|\bar{z}_t\|^2.
\end{aligned}
$$

By substituting the above inequality and (33) into (32), we obtain

$$
\begin{aligned}
& \mathbb{E}[\|\bar{u}_{t+1} - \nabla^2_{xy} g(\bar{x}_{t+1}, y_{t+1}^\star)\|_F^2] \\
& \leq (1 - \beta_t) \mathbb{E}[\|\bar{u}_t - \nabla^2_{xy} g(\bar{x}_t, y_t^\star)\|_F^2] + \frac{2\beta_t^2 \sigma_g^2}{K} + \frac{7\alpha_t^2 \widetilde{L}_g^2 (1 + L_y^2)}{2\beta_t} \mathbb{E}[\|\bar{z}_t\|^2] \\
& \quad + \frac{2\beta_t \widetilde{L}_g^2}{K} \sum_{k \in \mathcal{K}} \left( \|x_t^k - \bar{x}_{t+1}\|^2 + \|y_t^k - y_{t+1}^\star\|^2 \right).
\end{aligned}
$$

The desired result can be acquired by applying (28). This completes the proof. $\qquad\square$

## B.7 Proof of Theorem 5.1

*Proof:* We start our analysis by considering the term $\|\bar{y}_t - y^\star(\bar{x}_t)\|^2$. By rearranging (15), we have

$$
\begin{aligned}
\mathbb{E}[\|\bar{y}_t - y^\star(\bar{x}_t)\|^2] \leq{} & \frac{1}{\gamma_t \mu_g} \mathbb{E} \left( \|\bar{y}_t - y^\star(\bar{x}_t)\|^2 - \|\bar{y}_{t+1} - y^\star(\bar{x}_{t+1})\|^2 \right) \\
& + \mathcal{O} \left( \frac{\alpha_t^2}{(1 - \rho)^2} \right) + \frac{2\gamma_t \sigma_g^2}{\mu_g K} + \frac{3 L_y^2 \alpha_t^2}{\gamma_t^2 \mu_g^2} \mathbb{E}[\|\bar{z}_t\|^2].
\end{aligned}
\tag{34}
$$

Letting $m_1 = 4, m_2 = 12 L_g^2 L_q^2, m_3 = 12 C_f^2 L_q^2, m_{4,j} = 12 C_f^2 (1 - \kappa_g)^{b-j} L_g^{-2} \kappa_g^{-1}$, and $m_5 = 6 \left( L_f^2 (m_1 + m_2) + \widetilde{L}_g^2 (m_3 + m_4) \right)$ be the constants defined within (7), we define a random variable

$$
\begin{aligned}
P_t ={} & \frac{2}{\alpha_t} F(\bar{x}_t) + \frac{m_1}{\beta_t} \|\bar{s}_t - \nabla_x f(\bar{x}_t, y_t^\star)\|^2 + \frac{m_2}{\beta_t} \|\bar{h}_t - \nabla_y f(\bar{x}_t, y^\star(x_t))\|^2 \\
& + \frac{m_3}{\beta_t} \|\bar{u}_t - \nabla^2_{xy} g(\bar{x}_t, y^\star(x_t))\|_F^2 + \sum_{j=1}^b \frac{m_{4,j}}{\beta_t} \|\bar{v}_{t,j} - \nabla^2_{yy} g(\bar{x}_t, y^\star(x_t))\|_F^2.
\end{aligned}
$$

Here we observe that $P_0 \leq \mathcal{O} \left( \sqrt{\frac{T}{K}} \right)$ and $P_t \geq \frac{2}{\alpha_t} F(x^*) = 2 \sqrt{\frac{T}{K}} F(x^*)$. By multiplying $m_1 \beta_t^{-1}$, $m_2 \beta_t^{-1}, m_3 \beta_t^{-1}$, and $m_{4,j} \beta_t^{-1}$ to both sides of (25), (29), (30), and (31), respectively, and combining

with (11), we obtain that

$$\mathbb{E}[\|\nabla F(\bar{x}_t)\|^2] + \mathbb{E}[P_{t+1}]$$

$$\leq \mathbb{E}[P_t] + \mathcal{O}\left(\frac{\alpha_t^2 + \beta_t^2}{(1-\rho)^2}\right) + 6\big(L_f^2(m_1 + m_2) + L_g^2(m_3 + \sum_{j=1}^{b} m_{4,j})\big)\mathbb{E}[\|\bar{y}_t - y_t^\star\|^2]$$

$$+ \mathcal{O}\left(\frac{\beta_t}{K}\right) - \left(1 - \alpha_t L_F - \frac{4\alpha_t^2\big(L_f^2(m_1 + m_2) + \widetilde{L}_g^2(m_3 + \sum_{j=1}^{b} m_{4,j})\big)(1 + L_y^2)}{\beta_t^2}\right)\mathbb{E}[\|\bar{z}_t\|^2].$$

Letting $m_4 = 12C_f^2 L_g^{-2}\kappa_g^{-2}$, we note that $m_4 \geq \sum_{j=1}^{b} m_{4,j}$, and express the above inequality as

$$\mathbb{E}[\|\nabla F(\bar{x}_t)\|^2] + \mathbb{E}[P_{t+1}]$$

$$\leq \mathbb{E}[P_t] + \mathcal{O}\left(\frac{\alpha_t^2 + \beta_t^2}{(1-\rho)^2}\right) + 6\big(L_f^2(m_1 + m_2) + \widetilde{L}_g^2(m_3 + m_4)\big)\mathbb{E}[\|\bar{y}_t - y_t^\star\|^2]$$

$$+ \mathcal{O}\left(\frac{\beta_t}{K}\right) - \left(1 - \alpha_t L_F - \frac{4\alpha_t^2\big(L_f^2(m_1 + m_2) + \widetilde{L}_g^2(m_3 + m_4)\big)(1 + L_y^2)}{\beta_t^2}\right)\mathbb{E}[\|\bar{z}_t\|^2].$$

By multiplying $m_5$ to both sides of (34), combining the above inequality with (34), and recalling that $\alpha_t = C_0\sqrt{\frac{K}{T}}$ and $\beta_t = \gamma_t = \sqrt{\frac{K}{T}}$, we further obtain

$$\mathbb{E}[\|\nabla F(\bar{x}_t)\|^2] + \mathbb{E}[P_{t+1}] + \frac{m_5}{\mu_g}\sqrt{\frac{T}{K}}\mathbb{E}[\|\bar{y}_{t+1} - y_{t+1}^\star\|^2]$$

$$\leq \mathbb{E}[P_t] + \frac{m_5}{\mu_g}\sqrt{\frac{T}{K}}\mathbb{E}[\|\bar{y}_t - y_t^\star\|^2] + \mathcal{O}\left(\frac{K}{T(1-\rho)^2}\right) + \mathcal{O}\left(\frac{1}{\sqrt{KT}}\right)$$

$$- \underbrace{\left(1 - \frac{C_0 L_F}{\sqrt{T}} - 4C_0^2(L_f^2(m_1 + m_2) + \widetilde{L}_g^2(m_3 + m_4))(1 + L_y^2) - \frac{3m_5 L_y^2 C_0^2}{\mu_g^2}\right)}_{\Upsilon(C_0, T)}\mathbb{E}[\|\bar{z}_t\|^2].$$

We then observe that for large $T$, there exists a small constant $\tilde{C}_0 > 0$ such that $\Upsilon(C_0, T) \geq 0$ for all $C_0 \leq \tilde{C}_0$. In such scenario, we sum the above inequality over $t = 0, 1, \cdots, T-1$ and conclude that

$$\frac{1}{T}\sum_{t=0}^{T-1}\mathbb{E}[\|\nabla F(\bar{x}_t)\|^2] \leq \frac{P_0 + \frac{m_5}{\gamma_t \mu_g}\|\bar{y}_0 - y_0^*\|^2 - \mathbb{E}[P_T]}{T} + \mathcal{O}\left(\frac{1}{\sqrt{TK}}\right) + \mathcal{O}\left(\frac{1}{T(1-\rho)^2}\right)$$

$$\leq \mathcal{O}\left(\frac{1}{\sqrt{TK}}\right) + \mathcal{O}\left(\frac{K}{T(1-\rho)^2}\right),$$

where the last inequality applies the facts $P_0 + \frac{m_5}{\mu_g}\sqrt{\frac{T}{K}}\|\bar{y}_0 - y_0^*\|^2 \leq \mathcal{O}(\sqrt{\frac{T}{K}})$ and $P_T \geq 2\sqrt{\frac{T}{K}}F(x^*)$. This completes the proof. $\qquad\square$

## C   Proof of Results for $\mu$-PL Objectives

Throughout this subsection, we assume Assumptions 3.1, 3.2, 3.3, 3.4, and 5.2 hold. We set $b = 3\lceil\log_{\frac{1}{1-\kappa_g}} T\rceil$, consider the scenario where step-sizes follow (6) such that

$$\alpha_t = \frac{2}{\mu(C_1 + t)}, \text{ and } \beta_t = \gamma_t = \frac{C_1}{C_1 + t}, \quad \text{for } 1 \leq t \leq T,$$

where $C_1 > 0$ is a large constant making

$$\Psi(C_1) = \frac{1}{2} - \frac{2L_F}{\mu C_1} - \frac{8\alpha_t(L_f^2(z_1 + z_3) + \widetilde{L}_g^2(z_3 + z_4))(1 + L_y^2)}{\mu C_1} - \frac{6z_5 L_y^2}{\mu_g \mu C_1} > 0$$

with

$$z_1 = \frac{4}{\mu(C_1 - 2)}, \quad z_2 = \frac{12L_g^2 L_q^2}{\mu(C_1 - 2)}, z_3 = \frac{12C_f^2 L_q^2}{\mu(C_1 - 2)},$$

$$z_4 = \frac{12C_f^2}{\mu(C_1 - 2)L_g^2\kappa_g^2}, \quad \text{and} \quad z_5 = \frac{6C_1\big(L_f^2(z_1 + z_2) + \widetilde{L}_g^2(z_3 + z_4)\big)}{\mu_g(C_1 - 2/\mu)}. \tag{35}$$

## C.1 Lemma C.1 and Its Proof

**Lemma C.1** *Suppose Assumptions 3.1, 3.2, 3.3, 3.4, and 5.2 hold and the objective $F$ satisfies $\mu$-PL Assumption 5.2 in addition. We have*

$$\mathbb{E}[F(\bar{x}_{t+1})] - F^*$$
$$\leq \big(1 - \alpha_t\mu\big)\mathbb{E}[F(\bar{x}_t) - F^*] - \frac{\alpha_t}{2}(1 - \alpha_t L_F)\mathbb{E}[\|\bar{z}_t\|^2] + 2\alpha_t\mathbb{E}[\|\nabla_x f(\bar{x}_t, y_t^\star) - \bar{s}_t\|^2]$$
$$+ 6\alpha_t L_g^2 L_q^2 \mathbb{E}[\|\nabla_y f(\bar{x}_t, y_t^\star) - \bar{h}_t\|^2] + 6\alpha_t C_f^2 L_q^2 \mathbb{E}[\|\nabla_{xy}^2 g(\bar{x}_t, y_t^\star) - \bar{u}_t\|_F^2]$$
$$+ \frac{6C_f^2}{L_g^2\kappa_g} \sum_{j=1}^{b}(1 - \kappa_g)^{b-j}\mathbb{E}[\|\nabla_{yy}^2 g(\bar{x}_t, y_t^\star) - \bar{v}_{t,j}\|_F^2] + \mathcal{O}\left(\frac{\alpha_t\beta_t^2}{(1 - \rho)^2}\right).$$

*Proof:* Suppose the objective function $F$ satisfies the $\mu$-PL condition (5.2), by combining it with (13), we have

$$\mathbb{E}[F(\bar{x}_{t+1})] - \mathbb{E}[F(\bar{x}_t)]$$
$$\leq -\alpha_t\mu\mathbb{E}[F(\bar{x}_t) - F^*] - \frac{\alpha_t}{2}(1 - \alpha_t L_F)\mathbb{E}[\|\bar{z}_t\|^2]$$
$$+ \frac{\alpha_t}{K}\sum_{k=1}^{K}\left(\mathbb{E}[\|\nabla_x f(\bar{x}_t, y_t^\star) - s_t^k\|^2] + 3C_f^2 L_q^2 \mathbb{E}[\|\nabla_{xy}^2 g(\bar{x}_t, y_t^\star) - u_t^k\|^2]\right)$$
$$+ \frac{3\alpha_t}{K}\sum_{k=1}^{K}\left(L_g^2 C_f^2 \mathbb{E}[\|[\nabla_{yy}^2 g(\bar{x}_t, y_t^\star)]^{-1} - q_t^k\|^2] + C_g^2 L_q^2 \mathbb{E}[\|\nabla_y f(\bar{x}_t, y_t^\star) - h_t^k\|^2]\right).$$

By following (14) and applying the convergence of consensus errors in Lemma B.2, we further express the above inequality as

$$\mathbb{E}[F(\bar{x}_{t+1})] - \mathbb{E}[F(\bar{x}_t)]$$
$$\leq -\alpha_t\mu\mathbb{E}[F(\bar{x}_t) - F^*] - \frac{\alpha_t}{2}(1 - \alpha_t L_F)\mathbb{E}[\|\bar{z}_t\|^2] + 2\alpha_t\mathbb{E}[\|\nabla_x f(\bar{x}_t, y_t^\star) - \bar{s}_t\|^2]$$
$$+ 6\alpha_t L_g^2 L_q^2 \mathbb{E}[\|\nabla_y f(\bar{x}_t, y_t^\star) - \bar{h}_t\|^2] + 6\alpha_t C_f^2 L_q^2 \mathbb{E}[\|\nabla_{xy}^2 g(\bar{x}_t, y_t^\star) - \bar{u}_t\|_F^2]$$
$$+ \frac{6C_f^2}{L_g^2\kappa_g} \sum_{j=1}^{b}(1 - \kappa_g)^{b-j}\mathbb{E}[\|\nabla_{yy}^2 g(\bar{x}_t, y_t^\star) - \bar{v}_{t,j}\|_F^2] + \mathcal{O}\left(\frac{\alpha_t\beta_t^2}{(1 - \rho)^2}\right).$$

We acquire the desired result by substracting $F^*$ on both sides of the above inequality.

$\square$

## C.2 Proof of Theorem 5.3

Before establishing the convergence rate for $\mu$-PL function, we provide a result (Ghadimi and Lan, 2016, Lemma 1) to characterize the convergence behavior for a random sequence satisfying a special form of stochastic recursion as follows.

**Lemma C.2** *Letting $b_k = \frac{2}{k+1}$ and $\Gamma_k = \frac{2}{k(k+1)}$ for $k \geq 1$ be two nonnegative sequences. For any nonnegative sequences $\{A_k\}$ and $\{B_k\}$ satisfying*

$$A_k \leq (1 - b_k)A_{k-1} + B_k, \quad \text{for } k \geq 1,$$

*we have $\Gamma_k = \Gamma_s \prod_{j=s+1}^k (1 - b_j)$ and*

$$A_k \le \frac{\Gamma_k}{\Gamma_s} A_s + \sum_{i=s+1}^k \frac{\Gamma_k B_i}{\Gamma_i}.$$

We then derive the convergence rate of $\{\bar{x}_t\}$ for $\mu$-PL objectives as follows.

*Proof of Theorem 5.3:* First of all, under the choice of step-sizes that $\alpha_t = \frac{2}{\mu(C_1+t)}$ and $\beta_t = \gamma_t = \frac{C_1}{C_1+t}$, we have $\lim_{t\to\infty} \alpha_t = 0$, $\lim_{t\to\infty} \beta_t = 0$, and $\lim_{t\to\infty} \gamma_t = 0$. By following the analysis of Lemmas B.4, B.5, and B.6 and applying the convergence rates of consensus errors in Lemma B.2 (ii), we observe that (15), (25), (29), (30), and (31) still hold under this choice of step-sizes.

Next, we define a random variable

$$J_k = F(\bar{x}_t) - F^* + z_1 \|\nabla_x f(\bar{x}_t, y_t^\star) - \bar{s}_t\|^2 + z_2 \|\nabla_y f(\bar{x}_t, y_t^\star) - \bar{h}_t\|^2$$
$$+ z_3 \|\nabla_{xy}^2 g(\bar{x}_t, y_t^\star) - \bar{u}_t\|_F^2 + \sum_{1 \le j \le b} z_{4,j} \|\nabla_{yy}^2 g(\bar{x}_t, y_t^\star) - \bar{v}_{t,j}\|_F^2,$$

where

$$z_1 = \frac{2\alpha_t}{\beta_t - \alpha_t \mu} = \frac{4}{\mu(C_1 - 2)}, \quad z_2 = \frac{6\alpha_t L_g^2 L_q^2}{\beta_t - \alpha_t \mu} = \frac{12 L_g^2 L_q^2}{\mu(C_1 - 2)},$$
$$z_3 = \frac{6\alpha_t C_f^2 L_q^2}{\beta_t - \alpha_t \mu} = \frac{12 C_f^2 L_q^2}{\mu(C_1 - 2)}, \quad \text{and } z_{4,j} = \frac{6\alpha_t L_g^2 C_f^2 (1 - \kappa_g)^{b-j}}{(\beta_t - \alpha_t \mu) L_g^4 \kappa_g} = \frac{12 L_g^2 C_f^2 (1 - \kappa_g)^{b-j}}{\mu(C_1 - 2) L_g^4 \kappa_g},$$

are all constants defined in (35). By multiplying $z_1, z_2, z_3,$ and $z_{4,j}$ to both sides of (25), (29), (30), and (31), respectively, and combining them with Lemma C.1, we obtain

$$\mathbb{E}[J_{t+1}] \le (1 - \alpha_t \mu) \mathbb{E}[J_t] + \mathcal{O}\left(\frac{\beta_t^2}{K}\right) + \mathcal{O}\left(\frac{1}{t^3(1-\rho)^2}\right)$$
$$- \alpha_t \left(\frac{1 - \alpha_t L_F}{2} - \frac{7\alpha_t\left(L_f^2(z_1 + z_3) + L_g^2(z_3 + \sum_{1 \le j \le b} z_{4,j})\right)(1 + L_y^2)}{2\beta_t}\right) \mathbb{E}[\|\bar{z}_t\|^2]$$
$$+ 6\beta_t\left(L_f^2(z_1 + z_2) + L_g^2(z_3 + \sum_{1 \le j \le b} z_{4,j})\right) \mathbb{E}[\|\bar{y}_t - y_t^\star\|^2].$$

By noting that $\sum_{1 \le j \le b} z_{4,j} \le L_g^{-4} \kappa_g^{-2} = z_4$ in (35), we further express the above inequality as

$$\mathbb{E}[J_{t+1}] \le (1 - \alpha_t \mu) \mathbb{E}[J_t] + \mathcal{O}\left(\frac{\beta_t^2}{K}\right) + \mathcal{O}\left(\frac{1}{t^3(1-\rho)^2}\right)$$
$$- \alpha_t \left(\frac{1 - \alpha_t L_F}{2} - \frac{4\alpha_t\left(L_f^2(z_1 + z_3) + L_g^2(z_3 + z_4)\right)(1 + L_y^2)}{\beta_t}\right) \mathbb{E}[\|\bar{z}_t\|^2]$$
$$+ 6\beta_t\left(L_f^2(z_1 + z_2) + L_g^2(z_3 + z_4)\right) \mathbb{E}[\|\bar{y}_t - y_t^\star\|^2].$$

By recalling that $\alpha_t = \frac{2}{\mu(C_1+t)}$ and $\beta_t = \gamma_t = \frac{C_1}{C_1+t}$, we have $\alpha_t/\beta_t = \alpha_t/\gamma_t = \frac{2}{\mu C_1}$. Further, letting

$$z_5 = \frac{6\beta_t\left(L_f^2(z_1 + z_2) + L_g^2(z_3 + z_4)\right)}{\mu_g(\gamma_t - \alpha_t)} = \frac{6C_1\left(L_f^2(z_1 + z_2) + L_g^2(z_3 + z_4)\right)}{\mu_g(C_1 - 2/\mu)},$$

by multiplying $z_5$ to both sides of (15) and combining with the above inequality, we have

$$\mathbb{E}[J_{t+1}] + z_5 \mathbb{E}[\|\bar{y}_{t+1} - y_{t+1}^\star\|^2]$$
$$\le \left(1 - \frac{2}{C_1+t+1}\right)\left(\mathbb{E}[J_t] + z_5 \mathbb{E}[\|\bar{y}_t - y_t^\star\|^2]\right) + \mathcal{O}\left(\frac{\beta_t^2}{K}\right) + \mathcal{O}\left(\frac{1}{t^3(1-\rho)^2}\right)$$
$$- \alpha_t \underbrace{\left(\frac{1}{2} - \frac{2L_F}{\mu C_1} - \frac{8(L_f^2(z_1 + z_3) + L_g^2(z_3 + z_4))(1 + L_y^2)}{\mu C_1} - \frac{6z_5 L_y^2}{\mu_g \mu C_1}\right)}_{\Psi(C_1)} \mathbb{E}[\|\bar{z}_t\|^2].$$

We note that $\Psi(C_1)$ is increasing in $C_1$ with $\lim_{C_1 \to \infty} \Psi(C_1) = \frac{1}{2}$. Clearly, there exists a constant $\tilde{C}_1 > 0$ such that $\Psi(C_1) \geq 0$ for $C_1 \geq \tilde{C}_1$. Finally, we conclude that for $t \geq 0$,

$$\mathbb{E}[J_{t+1}] + z_5 \mathbb{E}[\|\bar{y}_{t+1} - y_{t+1}^\star\|^2]$$

$$\leq \left(1 - \frac{2}{C_1 + t + 1}\right)\left(\mathbb{E}[J_t] + z_5 \mathbb{E}[\|\bar{y}_t - y_t^\star\|^2]\right) + \mathcal{O}\left(\frac{\beta_t^2}{K}\right) + \mathcal{O}\left(\frac{1}{t^3(1-\rho)^2}\right)$$

$$\leq \frac{\Gamma_{C_1+t}}{\Gamma_{C_1}}[J_0 + z_5\|\bar{y}_0 - y_0^*\|^2] + \sum_{j=C_1}^{C_1+t} \mathcal{O}\left(\frac{\beta_j^2 \Gamma_{C_1+t}}{K\Gamma_j}\right) + \sum_{j=C_1}^{t+C_1} \mathcal{O}\left(\frac{\Gamma_{t+C_1}}{j^3(1-\rho)^2\Gamma_j}\right)$$

$$= \frac{C_1(C_1+1)J_0}{(C_1+t)(C_1+t+1)} + \frac{2}{(C_1+t)(C_1+t+1)}\left(\sum_{j=C_1}^{C_1+t} \mathcal{O}\left(\frac{j(j+1)}{j^2 K}\right) + \sum_{j=C_1}^{C_1+t} \mathcal{O}\left(\frac{j(j+1)}{j^3(1-\rho)^2}\right)\right)$$

$$\leq \frac{C_1(C_1+1)J_0}{(C_1+t)(C_1+t+1)} + \mathcal{O}\left(\frac{t}{(C_1+t)(C_1+t+1)K}\right) + \mathcal{O}\left(\frac{2\ln(C_1+t)}{(C_1+t)(C_1+t+1)(1-\rho)^2}\right)$$

$$\leq \mathcal{O}\left(\frac{1}{(t+1)K}\right) + \mathcal{O}\left(\frac{\ln t}{t^2(1-\rho)^2}\right),$$

where the second inequality uses Lemma C.2. This completes the proof.

$\square$

### C.3 Application to Distributed Risk-Averse Optimization

Letting $U^k(x, \xi^k)$ be a random utility function for agent $k \in \mathcal{K}$, we denote by $U(x) = \frac{1}{K}\sum_{k\in\mathcal{K}} \mathbb{E}_{\xi^k}[U^k(x,\xi^k)]$ the expected utility function averaged over all agents, and consider the following distributed regularized mean-deviation risk-averse optimization problem that

$$\max_x \left\{ U(x) - \kappa\left[\mathbb{E}_{\xi^1,\cdots,\xi^K}\left[U(x) - \frac{1}{K}\sum_{k\in\mathcal{K}} U(x^k, \xi^k)\right]_+^p\right]^{1/p} - \frac{\lambda\|x\|^2}{2}\right\}.$$

Here, all agents are connected by a decentralized network and work together to solve a shared risk-averse mean-deviation optimization problem (Ruszczyński and Shapiro, 2006). This problem can be reformulated as a three-level optimization problem that

$$\max_x f\left(x, y_1^*(x), y_2^*(x)\right) = y_1^*(x) - \kappa\left(y_2^*(x)\right)^{1/p} - \frac{\lambda\|x\|^2}{2},$$

$$\text{s.t. } y_1^*(x) = \underset{y_1 \in \mathbb{R}}{\text{argmin}} \, \mathbb{E}_{\xi^1,\cdots,\xi^K}\left(y_1 - \frac{1}{K}\sum_{k\in\mathcal{K}} U^k(x,\xi^k)\right)^2,$$

$$y_2^*(x) = \underset{y_2 \in \mathbb{R}}{\text{argmin}} \, \mathbb{E}_{\xi^1,\cdots,\xi^K}\left(y_2 - \left(y_1^*(x) - \frac{1}{K}\sum_{k\in\mathcal{K}} U^k(x,\xi^k)\right)_+^p\right)^2,$$

where $p > 1$ is a constant. The above problem is $\lambda$-strongly concave for any $\kappa \in (0, 1]$ and $\lambda > 0$ (Ruszczyński and Shapiro, 2006).

## D   Result for Single-center-multi-user-federated Stochastic Bilevel Optimization

We first provide the following Federated Sochastic Bilevel Optimization algorithm to tackle federated SBO over a Single-center-multi-user star network. Our algorithm utilizes the star network structure that a central server connects all agents and synchronously passes the same solution $(x_t, y_t)$ to all agents within each learning round. Then, each agent would query the stochastic information from its own data and pass it to the central server. After collecting all stochastic information, the central server would employ a similar stochastic approximation scheme as DSBO to update its estimation of the gradients and hessians within (3). The details are provided in Algorithm 2.

---

**Algorithm 2** Federated Stochastic Bilevel Optimization over Star Network

---

**input:** Step-sizes $\{\alpha_t\}$, $\{\beta_t\}$, $\{\gamma_t\}$, total iterations $T$.
    $x_0^k = \mathbf{0}$, $y_0^k = \mathbf{0}$, $u_0^k = \mathbf{0}$, $v_0^k = \mu_g \mathbf{I}$, $s_0^k = \mathbf{0}$, $h_0^k = \mathbf{0}$
1: **for** $t = 0, 1, \cdots, T-1$ **do**
2:    Communication: Pass $(x_t, y_t)$ to each agent $k \in \mathcal{K}$.
3:    **for** $k = 1, \cdots, K$ **do**
4:        Local sampling: Query $\mathcal{SO}$ at $(x_t, y_t)$ to obtain $\nabla_x f^k(x_t, y_t; \zeta_t^k)$, $\nabla_y f^k(x_t, y_t; \zeta_t^k)$,
                $\nabla_{xy}^2 g^k(x_t, y_t; \xi_t^k)$, and $\nabla_{yy}^2 g^k(x_t, y_t; \xi_t^k)$.
5:    **end for**
6:    Communication: Receive stochastic samples from each agent $k \in cK$.
7:    Outer loop update: $x_{t+1} = x_t - \alpha_t \left(s_t - u_t[v_t]^{-1} h_t\right)$.
8:    Inner loop update: $y_{t+1} = y_t - \frac{\gamma_t}{K} \sum_{k \in \mathcal{K}} \nabla_y g^k(x_t^k, y_t^k; \xi_t^k)$.
9:    Estimate $\nabla_x f(x_t, y_t)$: $s_{t+1} = (1 - \beta_t) s_t + \frac{\beta_t}{K} \sum_{k \in \mathcal{K}} \nabla_x f^k(x_t, y_t; \zeta_t^k)$.
10:   Estimate $\nabla_y f(x_t, y_t)$: $h_{t+1} = (1 - \beta_t) h_t + \frac{\beta_t}{K} \sum_{k \in \mathcal{K}} \nabla_y f^k(x_t, y_t; \zeta_t^k)$.
11:   Estimate $\nabla_{xy}^2 g(x_t, y_t)$: $u_{t+1} = (1 - \beta_t) u_t + \frac{\beta_t}{K} \sum_{k \in \mathcal{K}} \nabla_{xy}^2 g^k(x_t, y_t; \xi_t^k)$.
12:   Estimate $[\nabla_{yy}^2 g(x_t, y_t)]^{-1}$: Set $Q_{t+1} = \mathbf{I}$.
13:   **for** $i = 1, \cdots, b$ **do**
14:      $v_{t+1,i} = (1 - \beta_t) v_{t,i} + \frac{\beta_t}{K} \sum_{k \in \mathcal{K}} \nabla_{yy}^2 g^k(x_t^k, y_t^k; \xi_{t,i}^k)$,
15:      $Q_{t+1,i} = \mathbf{I} + (\mathbf{I} - \frac{1}{L_g} v_{t+1,i}) Q_{t+1,i-1}$
16:   **end for**
17:   Set $q_{t+1} = \frac{1}{L_g} Q_{t+1,b}$
18: **end for**
**output:** $\{x_t\}_{t=0}^{T-1}$.

---

**Theorem D.1 (Single-center-multi-user-federated SBO)** *Suppose 3.1, 3.3, and 3.4 hold. Letting $\{x_t\}$ be the sequence generated by Algorithm 2 over a star network with step-sizes defined in (7), we have*

*Proof:* We observe that the key difference between Alg. 2 and Alg. 1 is that Alg. 2 considers an environment where each agent synchronously receives a common solution $(x_t, y_t)$ from the central agent so that the effect of consensus disappears. Consequently, with a slight abuse of notation that $(x_t, y_t) = (\bar{x}_t, \bar{y}_t)$. Further, we observe that all consensus error terms within (11), (15), (25), (29), (30), and (31) would disappear, leading to tightened characterizations of the overall estimation errors under the star-network. For instance, (11) could be tightened as

$$\mathbb{E}[\|\nabla F(\bar{x}_t)\|^2]$$
$$\leq \frac{2}{\alpha_t} \Big( \mathbb{E}[F(\bar{x}_t)] - \mathbb{E}[F(\bar{x}_{t+1})] \Big) - (1 - \alpha_t L_F) \mathbb{E}[\|\bar{z}_t\|^2] + 4 \mathbb{E}[\|\nabla_x f(\bar{x}_t, y_t^\star) - \bar{s}_t\|^2]$$
$$+ 12 L_g^2 L_q^2 \mathbb{E}[\|\nabla_y f(\bar{x}_t, y_t^\star) - \bar{h}_t\|^2] + 12 C_f^2 L_q^2 \mathbb{E}[\|\nabla_{xy}^2 g(\bar{x}_t, y_t^\star) - \bar{u}_t\|_F^2]$$
$$+ \frac{12 C_f^2}{L_g^2 \kappa_g} \sum_{1 \leq j \leq b} (1 - \kappa_g)^{b-j} \mathbb{E}[\|\nabla_{yy}^2 g(\bar{x}_t, y_t^\star) - \bar{v}_{t,i}\|_F^2].$$

By following the analysis of Theorem 5.1 and employing the step-sizes in (7), we obtain that

$$\frac{1}{T} \sum_{t=0}^{T-1} \|\nabla F(x_t)\|^2 \leq \mathcal{O}\left(\frac{1}{\sqrt{KT}}\right).$$

This completes the proof.

# E   Additional Numerical Experiments

## E.1  Hyper-parameter Optimization

The baseline algorithm DBSA conducts the followings. At the outer solution $x_t^k$, DBSA obtains an estimator $y_t^k$ of $y^\star(x_k)$ via conducting $t$ gossip stochastic gradient descent steps $\tilde{y}_{t,i+1}^k =$

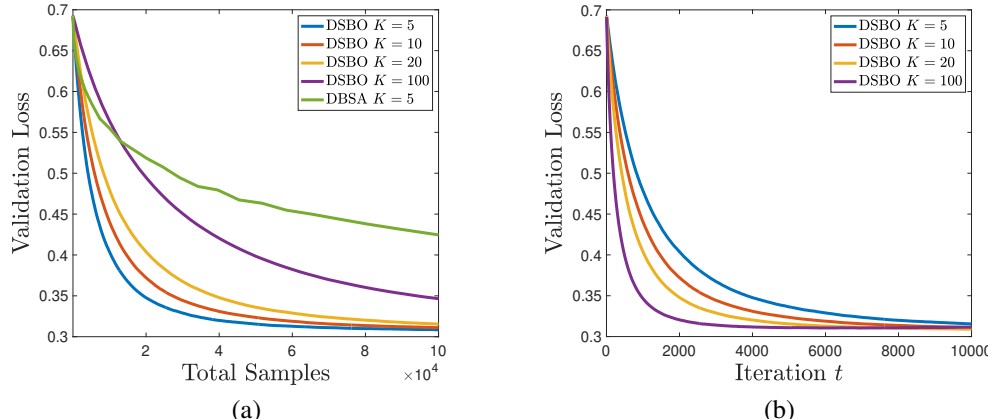

(a)                                        (b)

Figure 4: Performances of Algorithm 1 and DSBA over a uniform fully connected network: (a) Empirical averaged training loss against total samples for DSBO $K = 5, 10, 20, 100$ and DSGD $K = 5$ (b) Empirical averaged validation loss against iteration for DSBO $K = 5, 10, 20, 100$. All figures are generated through 10 independent simulations over australia handwriting dataset.

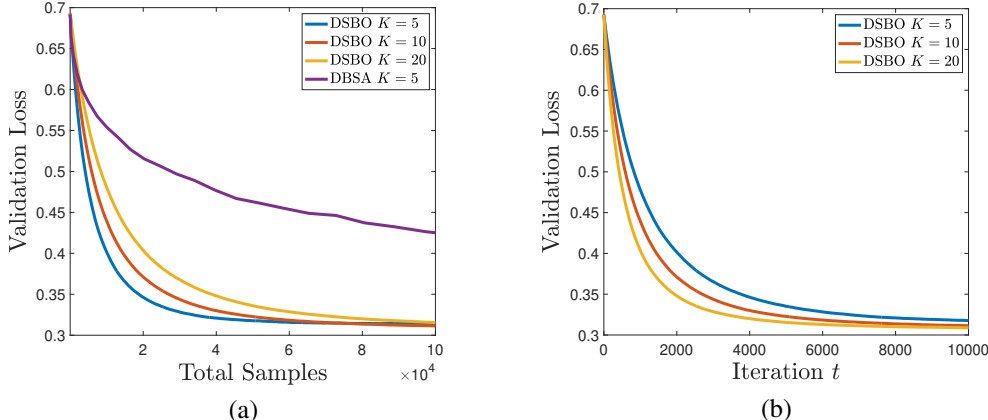

(a)                                        (b)

Figure 5: Performances of Algorithm 1 and DSBA over a connected random network: (a) Empirical averaged training loss against total samples for DSBO $K = 5, 10, 20$ and DBSA $K = 5$. (b) Empirical averaged validation loss against iteration for DSBO $K = 5, 10, 20$. All figures are generated through 10 independent simulations over australia handwriting dataset.

$\sum_{j \in \mathcal{N}_k} w_{k,j} \tilde{y}_{t,i}^j - \eta_{t,i} \nabla_y g(x_t^k, \tilde{y}_{t,i}^k; \xi_i^k)$ for $i = 0, 1, \cdots, t$ with $y_t^k = \tilde{y}_{t,t}^k$, and then update the main solution $x_t^k$ by one stochastic gradient descent step that $x_{t+1}^k = \sum_{j \in \mathcal{N}_k} w_{k,j} x_t^j - \alpha_t \nabla_x f(x_t^k, y_t^k; \zeta_t^k)$. We summarize the details in Algorithm 3.

We test our algorithm over various networks to further investigate its performance. Specifically, we test our algorithm over fully connected networks with $K = 5, 10, 20, 100$ and randomly generated connected networks with $K = 5, 10, 20$. We summarize the details in Figures 4 and 5, respectively. These numerical results suggest the efficiency and robustness of our algorithm over networks of large size and various topologies.

## E.2 Distributed Policy Evaluation for Reinforcement Learning

**Simulation environment:** In our experiments, for each state $s \in \mathcal{S}$, we generate its feature $\phi_s \sim$ Unif$[0,1]^m$; we uniformly generate the transition probabilities $p_{s,s'}$ and standardize them such that $\sum_{s' \in \mathcal{S}} p_{s,s'} = 1$; we sample the mean of rewards $\bar{r}_{s,s'}^k \sim$ Unif$[0,1]$ for all $s \in \mathcal{S}$ and each agent $k \in [K]$. We set the regularizer parameter $\lambda = 1$.

---

**Algorithm 3** Decentralized Bilevel Stochastic Approximation

---

**input:** Step-sizes $\{\alpha_t\}$, $\{\eta_{t,i}\}$, number of total iterations $T$.

$\quad x_0^k = \mathbf{0}, y_0^k = \mathbf{0}$

1: **for** $t = 0, 1, \cdots, T-1$ **do**
2: $\quad$ Inner loop update:
3: $\quad$ **for** $i = 0, 1, \cdots, t$ **do**
4: $\quad\quad$ **for** $k = 1, 2, \cdots, K$ **do**
5: $\quad\quad\quad$ Local sampling: Query $\mathcal{SO}$ at $(x_t^k, \tilde{y}_{t,i}^k)$ to obtain $\nabla_y g(x_t^k, \tilde{y}_{t,i}^k; \xi_{t,i}^k)$.
6: $\quad\quad\quad$ Estimate: $\tilde{y}_{t,i+1}^k = \sum_{j \in \mathcal{N}_k} w_{k,j} \tilde{y}_{t,i}^j - \eta_{t,i} \nabla_y g(x_t^k, \tilde{y}_{t,i}^k; \xi_{t,i}^k)$.
7: $\quad\quad$ **end for**
8: $\quad$ **end for**
9: $\quad$ Outer loop update: $x_{t+1}^k = \sum_{j \in \mathcal{N}_k} w_{k,j} x_t^j - \alpha_t \nabla_x f(x_t^k, y_t^k; \zeta_t^k)$.
10:
11: **end for**
**output:** $\bar{x}_T = \frac{1}{K} \sum_{k \in \mathcal{K}} x_T^k$.

---

---

**Algorithm 4** Decentralized Stochastic Gradient Desecent

---

**input:** Step-sizes $\{\alpha_t\}$, weights $\{\eta_{t,i}\}$, number of total iterations $T$.

$\quad x_0^k = \mathbf{0}, y_0^k = \mathbf{0}$

1: **for** $t = 0, 1, \cdots, T-1$ **do**
2: $\quad$ Inner value update: Set $\tilde{y}_{t,0}^k = 0$.
3: $\quad$ **for** $i = 0, 1, \cdots, t-1$ **do**
4: $\quad\quad$ **for** $k = 1, \cdots, K$ **do**
5: $\quad\quad\quad$ Local sampling: Query $\mathcal{SO}$ at $x_t^k$ to obtain $g^k(x_t^k; \xi_{t,i}^k)$.
6: $\quad\quad\quad$ Estimate: $\tilde{y}_{t,i+1}^k = (1 - \eta_{t,i}) \sum_{j \in \mathcal{N}_k} w_{k,j} \tilde{y}_{t,i}^j + \eta_{t,i} g^k(x_t^k; \xi_{t,i}^k)$.
7: $\quad\quad$ **end for**
8: $\quad$ **end for**Set $y_t^k = \tilde{y}_{t,t}^k$.
9: $\quad$ Outer loop update: $x_{t+1}^k = \sum_{j \in \mathcal{N}_k} w_{k,j} x_t^j - \alpha_t \nabla g(x_t^k; \xi_t^k) \nabla f(y_t^k; \zeta_t^k)$.
10:
11: **end for**
**output:** $\bar{x}_T = \frac{1}{K} \sum_{k \in \mathcal{K}} x_T^k$.

---

In each simulation, we set $|\mathcal{S}| = 100$ and update the solution $(x^k, y^k)$ for each agent in a parallel manner as follows: At iteration $t$, for each state $s \in \mathcal{S}$, we simulate a random transition to another state $s' \in \mathcal{S}$ using the transition probability $p_{s,s'}$'s, generate a random reward $r_{s,s'}^k \sim \mathcal{N}(\bar{r}_{s,s'}^k, 1)$, and update $x_t^k$ using step-sizes $\alpha_t = \min\{0.01, \frac{2}{\lambda t}\}$ and $\beta_t = \gamma_t = \min\{0.5, \frac{50}{t}\}$.

We provide the details of the baseline algorithm DSGD in Algorithm 4.

To further study the linear speedup effect under various accuracy levels, we compute the total generated samples for finding an $\epsilon$-optimal solution $\|\bar{x}_t - x^*\|^2 \leq \epsilon$ and plot the 75% confidence region of log-sample against number of agents $K = 5, 10, 20$ for various $\epsilon$'s in Figure 6. Similar as in Figure 2, we observe for all accuracy levels $\epsilon = 0.8 \times 10^{-6}, 1.5 \times 10^{-6}, 2 \times 10^{-6}$, the required samples for finding an $\epsilon$-optimal solution by $K$ agents are roughly the same, further demonstrating the linear speedup effect.

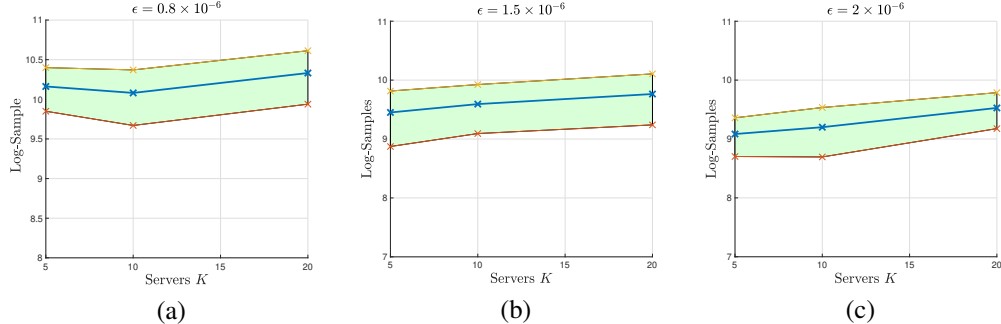

Figure 6: 75% confidence region of log- total samples for achieving $\|\bar{x}_t - x^*\|^2 \leq \epsilon$ with varying network sizes $K = 5, 10, 20$ for $\epsilon = 0.8 \times 10^{-6}, 1.5 \times 10^{-6}, 2 \times 10^{-6}$.