# OpenReview forum: "Decentralized Gossip-Based Stochastic Bilevel Optimization over Communication Networks"
_NeurIPS.cc/2022/Conference — NeurIPS 2022 Accept_

### Official Review · Reviewer_jXmE · 2022-06-29

**Rating:** 7
**Confidence:** 3
**Soundness:** 2 fair
**Presentation:** 3 good
**Contribution:** 2 fair

**Summary:**

This paper studies the distributed bilevel optimization over a network under the non-convex cases and strongly-convex cases. This paper first develops a gossip-based distributed bilevel learning algorithm based on the stochastic gradient descent method. Then, the authors develop the convergence analysis to characterize the convergence behavior of the stated algorithm. The algorithm achieves an $\mathcal{O}\left(\frac{1}{K \epsilon^{2}}\right)$ sample complexity for nonconvex objectives and achieves an $\mathcal{O}\left(\frac{1}{K \epsilon}\right)$ sample complexity for $\mu$-PL functions, subsuming strongly convex optimization.

**Questions:**

1. What is the definition of $F^*$ in Line. 229/237?
2. The convergence result for Theorem 1 is for the $\epsilon$-stationary point, while the result for Theorem 2 is for $\epsilon$-optimal point, why not make them being consistent?
3. How to determine the Matrix W in your experiment? How does the network sparsity affect the Matrix W and the convergence result?
4. Are there any motivational examples of performing the decentralized bilevel optimization problems with strongly-convex objectives?
5. The Distributed policy evaluation for reinforcement learning shown in Section.6 shows in Line.285 that

$y_s^*(x) =\phi_{s}^{\top} x-\mathbb{E}_{ s' } [ 1/K \sum_k r^{k}( s , s' ) +\gamma  \phi_s^{\top}  x  |  s] $

However,  $ y^{*}(x)=\underset{y \in \mathbb{R}^{d_{y}}}{\operatorname{argmin}} \frac{1}{K} \sum_{k=1}^{K} g^{k}(x, y)$ in eqs.(1).

What is the g function in Line 285? In eqs(1), should the $y^*$ be the solution when optimizing the lower-level $g$ function, but not the direct result of a function?

6.  In Line.45, it shows that $x$ is a vector of hyper-parameters for a strongly convex regularizer $R$ and the loss function $l_i(y)$ only depends on $y$. However, eqs.(1) shows that the loss function includes both $x$ and $y^*(x)$.
 The question is: how to calculated $\nabla_x l_{i}( y^*(x) )$ shown in Eqs.(3) without x inside? Is $\nabla_x l_{i}( y^*(x) )$=0 ?The stated hyperparameter problem seems different from the bilevel optimization problem in (1).








**Strengths And Weaknesses:**

Strengths:

This paper presents a clear problem formulation for the decentralized bilevel optimization problem motivated by several real-world applications. Besides, the proposed gossip-based distributed bilevel learning algorithm and experimental evaluations are mostly well explained.

Weaknesses:
1. This submission missed some important theoretical results of existing works. The state-of-the-art sample complexities (paper shown in Line.111 and Line.112) for the stochastic bilevel optimization problem is $\mathcal{O}(\epsilon^{-1.5})$, while the authors claim that the state-of-the-art sample complexity is  $\mathcal{O}(\epsilon^{-2})$.

2. The stochastic estimator seems infeasible. In machine learning applications,  $\nabla_{yx}^2 g$ is usually within a high-dimensional. The question is, in Algorithm 1, Line 4, how to compute the Hessian inverse with a relatively high-dimensional matrix?

3. Theoretical results: Even though this paper is the first paper to explore the decentralized bilevel optimization problem. The proof techniques seem regular, and the results in both nonconvex and strongly convex cases are not attractive. First, in the case of the nonconvex objectives, the sample complexity of $\mathcal{O}(\epsilon^{-2})$ per client is not a surprising result, and the state-of-the-art result (single-client bilevel optimization problem) already achieves the sample complexity of $\mathcal{O}(\epsilon^{-1.5})$. Second, the strongly-convex objectives are not widely used in real-world applications. It would be helpful if the authors could provide some motivational examples to support the reasons for further exploring the strongly convex decentralized bilevel optimization problem.

4. The experimental evaluations are weak. First, the Australia handwriting dataset and LR models are too simple. Evaluating the optimization algorithms using CIFAR10/ImageNet and deeper neural networks has become a standard requirement. Second, it would be better for the authors to provide additional experiments on different distributed network topologies (dense/ sparse). Third, the major contribution of this paper is to provide the convergence bound of the gradient norm. The authors should also provide the gradient norm comparison in the experiment section.




Minors:
1.Line 96: sammple --> sample

---

> ### Author Response · Authors · 2022-08-02
> **Responses to Reviewer jXmE**
>
> **Question.** The state-of-the-art sample complexities (paper shown in Line.111 and Line.112) for the stochastic bilevel optimization problem is $O(1/\epsilon^{1.5})$ , while the authors claim that the state-of-the-art sample complexity is $O(1/\epsilon^{2})$ .
> \
> **Response.**
> We would like to point out that the two settings are quite different. The algorithms that achieve $O(1/\epsilon^{1.5})$ sample complexities are rooted in momentum [1] or variance reduction [2], which requires using samples that are not available in our setting. For instance, the SVRB algorithm of [2] employs the momentum technique by using $\nabla_y f(x_t,y_t;\xi_t)$  and $\nabla_y f(x_{t-1},y_{t-1};\xi_t)$ in iteration $t$ - **obtaining such samples is not possible with a typical stochastic gradient oracle**. In our setting (analogous to the typical SGD setting), all sampled gradients/hessians are independently queried. We can get one noisy gradient per query, but we can never have both
>  $\nabla_y f(x_t,y_t;\xi_t)$  and $\nabla_y f(x_{t-1},y_{t-1};\xi_t)$ at the same time. In our setting, the best rate is $O(1/\epsilon^2)$, and it is well known to be unimprovable. We have added this reference to our paper.
>
> [1] Khanduri et al., A near-optimal algorithm for stochastic bilevel optimization via double-momentum.
> \
> [2] Guo et al., Randomized Stochastic Variance-Reduced Methods for Multi-Task Stochastic Bilevel Optimization.
>
> **Question.** Hessian inverse
> \
> **Response.**
> Good question! We agree that it is expensive to invert the Hessian for high-dim applications. In our revised manuscript, we have revised our algorithm to approximate the hessian inverse by using
>  $b = \Theta( \log(T))$ independent estimators $\\{v_{t,j}^k\\}_{j=1}^b$ of the hessian **(lines 9-14 of Algorithm 1 in the revised submission)**. This reduces the per-iteration computational cost from $O(d_y^3)$ to $O(\log(T)d_y^2)$, making the algorithm tractable for high-dimensional problems.
>
> **Question.** Strongly convexity.
> \
> **Response.**
> + First, for many convex problems, such as linear regression and logistics regression, letting $x_i$ be the feature associated with data point $(x_i,y_i)$, the objective is strongly convex within a unit ball around the optimal solution when $\mathbf{E}[x_i x_i^\top]$ is non-singular, where $\mathbf{E}$ is taken over the random feature $x_i$.
>
> + Second, in real-world machine learning applications, regularized approaches are often employed to avoid overfitting and many regularizers, including the $l_2$-norm  preserve strong convexity. Introducing such a regularizer would make a convex objective to be strongly convex. For instance, our convex MDP residual minimization problem becomes strongly convex in the presence of an $l_2$-regularizer $\lambda||x||^2$. As Theorem 2 suggests that our algorithm establishes a faster $O(1/T)$ convergence rate for strongly convex problems, our study would improve the algorithm efficiency in the presence of strongly-convex regularizers.
>
> + Third, we provide another strongly convex application, the risk-averse mean-deviation optimization, in Section C.3 of the supplement.
>
> **Question.** Numerical experiments.
> \
> **Response.** Thanks for your suggestions. In our revision, we have provided the following **additional experiments, reported in Figures 4, 5, 6 and Section E.1 of the revised supplement.**
> + We test our algorithm over networks of different topologies, including a fully-connected network ($K=5,10,20,100$) and a randomly connected network ($K=5,10,20$). We provide the details in Figures 4 and 5,  Section E.1 of the supplement, respectively. These results show that our algorithm converges rapidly under various network topologies and demonstrates its efficiency. Also, Figure 4 shows that our algorithm converges rapidly when $K = 100$, which demonstrates the efficiency of our algorithm for large networks.
> + We evaluate the average squared gradient norm of the solution path. We plot the logged squared grad norm $\log( \sum_{i=1}^t \|\nabla F(\bar x_i) \|^2 )$ against the log-iteration, $\log t$, in  Figure 6, Section E.1 of the supplement to validate our $\mathcal{O}(1/\sqrt{T})$  convergence rate result Theorem 1, with an additional  straight line of slope -0.5 provided for benchmark comparison.
> Figure 6 suggests that the slope of $\log(\sum_{i=1}^t||\nabla F(\bar x_i)||^2)$ against $\log t$ is around $-1/2$, which validates our $\mathcal{O}(1/\sqrt{T})$ convergence rate claim in Theorem 1.
>
> **Question.** Definition of $F^*$
> \
> **Response.** Here $F^*$ is defined as the optimal value for problem (1). We have provided this definition in line 35 of our revision.

---

> > ### Author Response · Authors · 2022-08-02
> > **Responses to Reviewer  jXmE (Continued)**
> >
> > **Question.** $\epsilon$-stationary point
> > \
> > **Response.**
> > Due to the non-convexity, our algorithm is not guaranteed to generate a sequence that converges to the optimal solution $x^*$, so that we are unable to find $\epsilon$-optimal points for nonconvex problems. It is worth mentioning the similar metric ($\epsilon$-stationary point) is also adopted by prior work in nonconvex bilevel optimization [1,2].
> >
> >
> > **Question.** Construction of $W$ and effect of network
> > \
> > **Response.**
> > (i) $W$ is a doubly-stochastic matrix that depends on the network topology. Specifically, we can assign a possible value $w_{i,j} = w_{j,i}>0$ if nodes $i$ and $j$ are connected, while we must have $w_{i,j} = w_{j,i} = 0$ otherwise.  The weights must also satisfy the constraints $\sum_{i}w_{i,j} = 1$ and $\sum_{j}w_{i,j} = 1$ for all nodes $i,j$. (ii) Briefly speaking, the convergence is slower for sparse networks than for dense networks because of the slower information diffusion. The effect of network topology is interpreted by the spectral gap $(1-\rho)$ of $W$, where $\rho = \lambda_{\max}(W - \frac{1}{K}\mathbf{1} \mathbf{1}^\top ) $. Specifically, our algorithm converges  in the rate  of $O(\frac{1}{\sqrt{KT}} + \frac{K}{ T(1-\rho)^2 })$ for nonconvex problems and converges in the rate of $O(\frac{1}{KT} + \frac{ \ln T}{  T^2(1-\rho)^2} )$ for strongly convex problems. It is worth emphasizing that terms associated with the spectral gap $(1-\rho)$ are dominated by $O(\frac{1}{ \sqrt{KT} } )$ and $O(\frac{1}{ KT } )$ for noncovnex and strongly convex problems, respectively, implying that the network topology would not affect the convergence rate when the number of iterations $T$ is large.
> >
> > **Question.** Expression of $g$ in Line 285
> > \
> > **Response.**
> > Sorry for the confusion. In this application, the lower-level function writes
> >  $$
> >  g(x,y)  = \mathbf{E}\_{s'} [(\phi_s^\top x -\frac{1}{K}\sum_k r^k(s,s')- y)^2],
> >  $$
> > so that $y^*(x)$ admits a closed-form expression that
> >  $$y^*(x) = \phi_s^\top x - \mathbf{E}_{s'} [  \frac{1}{K}\sum_k r^k(s,s') + \gamma \phi_s^\top x | s].
> >  $$
> >
> > **Question.** Is $\nabla_x l_i(y^*(x))=0$?
> > \
> > **Response.**
> >  You are right! In this hyper-parameter optimization application, the outer level targets minimizing the validation loss while the inner level aims at finding the optimal hyper-parameter $x \in \mathbf{R}^{d_x}$. In this case, $x$ only implicitly affects the outer level and we have
> > $\nabla_x l_i(y^*(x)) = 0$ because $l_i(y)$ only takes the vector $y \in \mathbf{R}^{d_y}$ as  input.

---

> > > ### Comment · Reviewer_jXmE · 2022-08-03
> > > **Reviewer's Comments**
> > >
> > > Thanks for your response and detailed explanation of my questions.
> > >
> > > Now I see your more contributions to the optimization area.
> > >
> > > I will raise the score to 7.

---

### Official Review · Reviewer_UN8Y · 2022-07-11

**Rating:** 6
**Confidence:** 5
**Soundness:** 3 good
**Presentation:** 3 good
**Contribution:** 3 good

**Summary:**

In this work, the authors studied the stochastic bilevel optimization problem in a decentralized way. The authors proposed a gossip-based distributed bilevel learning algorithm that solve both the inner and outer optimization problems in a single timescale. Furthermore, the sample complexities are optimal for $\epsilon$ and $K$, and the method achieves linearly scaling in network size. Although there are many existing works for the stochastic bilevel optimization, the successful work for decentralized SBO is still limited. This work solve the DSBO by estimating the $y^*(x)$ and handled the network propagation via gossip-based methods. The proposed method each agent solves an optimization problem collaboratively by sampling stochastic first- and second-order information using its data and making gossip communications with its neighbors. In addition, the theoretical convergence guarantees are also presented, which match the state-of-the-art results for non-federated stochastic bilevel optimization and central-server regime, and suggest that the algorithm exhibits the linear speed-up effect for decentralized settings. The method is verified by the experiments on decentralized ring networks.

**Questions:**

The work only consider the artificially constructed decentralized ring networks, I would suggest to evaluate the methods on multiple different networks to demonstrate the method effectiveness and scaling.

**Limitations:**

the authors adequately addressed the limitations and potential negative societal impact of their work

**Strengths And Weaknesses:**

The strength of the work is the theoretical analysis and proposed method for decentralized stochastic bilevel optimization problem. As I mentioned, although there are many existing works for the stochastic bilevel optimization, the successful work for decentralized SBO is still limited. This work solve the DSBO by estimating the $y^*(x)$ and handled the network propagation via gossip-based methods. In addition, the theoretical convergence guarantees are also presented, which match the state-of-the-art results for non-federated stochastic bilevel optimization and central-server regime, and suggest that the algorithm exhibits the linear speed-up effect for decentralized settings.

The weakness of the work is the experiments. The work only consider the artificially constructed decentralized ring networks, and only run two simple examples on hyper-parameter optimization and policy evaluation in Markov Decision Processes (MDP). I would suggest to evaluate the methods on multiple different networks to demonstrate the method effectiveness and scaling. Also suggesting more practical examples for validation purpose. For example, most experiments are running using only 5 nodes, far behind the practical decentralized setting, suggest nodes from 20 to 100, and network topology from ring network to many other variations.

---

> ### Author Response · Authors · 2022-08-02
> **Responses to Reviewer UN8Y**
>
> Thanks for your valuable and instructive comments, which have helped us improved our paper. Below we provide point-to-point responses to your comments.
>
> **Question.** Experiments.
> \
> **Response.**
> Thank you for the suggestions on numerical experiments. After revision, we have **added new experiments with a variety of networks, reported in Figures 4, 5, 6 and  Section E.1 of the revised supplement**. In short:
> + We test the performance of our algorithm over a large-scale fully-connected network consisting of 100 nodes. The results are provided in Figure 4 in Section E.1 of the supplement. Figure 4 shows that our algorithm converges rapidly when $K = 100$, which demonstrates the efficiency of our algorithm for large networks.
> + We test the performance of our algorithm over random connected networks. We provide the results in Figures 5 and 6 in Section E.1 of the supplement. Our algorithm efficiently finds convergent solutions under both network topologies, suggesting its efficiency and robustness.
>
>  We hope these could address your concerns.

---

### Official Review · Reviewer_s3u4 · 2022-07-11

**Rating:** 7
**Confidence:** 4
**Ethics Flag:** Yes
**Soundness:** 3 good
**Presentation:** 4 excellent
**Contribution:** 3 good

**Summary:**

The paper tackles the problem of Stochastic Bilevel Optimization (SBO) in a decentralized setting, using a gossip protocol for communication between the agents. The authors propose an approach based on a existing formulation of the gradient and a recent contribution (Ghadimi et al. 2020) to provide an (biased) estimate of the Hessian in a decentralized framework. They then provide a complete theoretical analysis of their method for both $\mu$-PL and nonconvex objectives. Finally, empirical results are provided for the two problems introduced in the beginning of the paper: hyperparameter optimization and policy evaluation.

**Questions:**

- Is there another way of performing decentralized estimation of the Hessian over a network? If so, how would it compare to the method used here?
- What is the motivation to only use ring networks in the experiments? These networks are often slow in terms of "diffusion" and are a counter-example to the robustness mentioned on l. 174.
- I do not understand the notation on l. 139, is $\nabla_y f^k$ only Lipschitz continuous on $x$? on $y$?

**Limitations:**

Privacy if briefly mentioned, although it does not seem adequate to me.

**Strengths And Weaknesses:**

Overall, the paper is very clear and easy to follow. SBO is thoroughly introduced and illustrated, and the main challenge of producing a decentralized approach (combining the recent formulation of the gradient with the Hessian estimation) clearly stated. Each step of the analysis is first motivated then commented, with an attention to providing easy-to-read quantities.
Empirical results are extensively commented and seem convincing about the validity of the approach.

Some remarks:
- Polyak-Łojasiewicz should be introduced at some point rather than be stated as "_PL_" from the start.
- l. 171: ensuring data privacy just based on the fact that raw data are not communicated is quite ambitious, to say the least.
- l. 174: it is true that the method would not collapse from a single failure in the network, but I think it is worth mentioning that the optimization objective would then be different, as the network would be missing a part of the dataset.

Typos:
- Step 9 of Algorithm 1: UnifoRm
-  Equation (3): should be $\nabla_x f$ rather than $\nabla f_x$

---

> ### Author Response · Authors · 2022-08-02
> **Responses to Reviewer s3u4**
>
> Thanks for your positive and valuable comments, which have helped us improve the paper. Below we provide point-to-point responses to your comments.
>
> **Question.** PL condition
> \
> **Response.** Thank you for this point. We have revised it line 95 accordingly.
>
> **Question.** Data privacy
> \
> **Response.**
>  We agree with you that not communicating the raw data is a naive approach to protecting privacy. In some applications such as healthcare, the hospitals are not allowed to share the raw data, so they are not able to aggregate the data from different hospitals and solve a shared target problem. In contrast, our approach provides a solution to cope with such a scenario. Further, it is worth emphasizing that for each agent, our algorithm only shares the stochastic gradients/Hessians with its neighbors. In the scenario that each agent has different trust levels toward the rest of the agents, it may only share this information with reliable agents to protect privacy, which results in a relatively sparse communication network. Our algorithm can also handle such a restriction if the network is connected.
>
> **Question.** Collapse of network
> \
> **Response.**
> We are sorry for the confusion caused by our previous argument. Indeed, the decentralized network is robust to contingencies in the network. If some communication channel fails, the agents can still jointly learn provided that the network is still connected, and the optimization objective remains unchanged. In contrast, a one-center-multi-agent system would become unconnected if the central server fails, which also leads to a change in the optimization objective. We have revised our manuscript accordingly.
>
> **Question.** Estimating inverse of hessian
> \
> **Response.**
> This is a good point worth emphasizing. To reduce the cost of inverting the hessian, we have refined our algorithm to approximate the inverse of the hessian by using $b = \Theta( \log(T))$ independent estimators $\\{ v_{t,j}^k \\}_{j=1}^b$ of the hessian (lines 9-14 of Algorithm 1). Both approaches enjoy the $\mathcal{O}(\frac{1}{\sqrt{T}})$ rate of convergence for nonconvex problems, and enjoy the $\mathcal{O}(\frac{1}{KT})$ rate of convergence for $\mu$-PL problems. The original approach suffers from a cost of $\mathcal{O}(d_y^3)$ in inverting the hessian within each iteration, while the per-iteration computation cost for our refined approach is $\mathcal{O}(\log(T) d_y^2)$, which is cheaper for high-dimensional problems.
>
> **Question.** Motivation for experiments on ring networks
> \
> **Response.**
> We test the ring networks for demonstrating the efficiency and robustness of our algorithm for applications with slow diffusion. For the ring networks, if one communication channel fails, the network remains connected so that our algorithm is still applicable.
>
> After revision, we have **included additional experiments with a variety of networks, reported in Figures 4, 5, 6, and Section E.1 of the revised supplement**.
>
> **Question.** Lipschitz continuity of $\nabla_y g^k(x,y)$
> \
> **Response.** Here we assume that these first- and second-order information are Lipschitz continuous in both $x$ and $y$ that for any $x_1,x_2 \in \mathbf{R}^{d_x}$ and $y_1,y_2 \in \mathbf{R}^{d_y}$,
> $$
> || \nabla_y g^k(x_1,y_1) - \nabla_y g^k(x_2,y_2) || \leq L_g( || x_1 - x_2|| + || y_1 + y_2||).
> $$
> We have provided this statement in our revised manuscript.
>
> **Question.** Typos.
> \
> **Response.** Thanks for pointing these out. We have corrected them in our revision.

---

### Official Review · Reviewer_8YY4 · 2022-07-11

**Rating:** 6
**Confidence:** 3
**Soundness:** 3 good
**Presentation:** 3 good
**Contribution:** 2 fair

**Summary:**

This paper considers the decentralized bilevel optimization problem. It proposes a gossip-based method which overcomes the challenges in computing the hyper-gradients in decentralized bilevel optimization method. With a strongly-convex lower level problem, the method is provably convergent and achieves the optimal rate (along with linear speedup) for both the general non-convex upper level problem and the PL upper level problem.

**Questions:**

1. As stated in previous section, why is Assumption A. 1. (iv), which is stronger than the standard bounded variance assumption, needed in the analysis? Is it possible to relax this assumption to the more standard one?

2. Can the authors explain more about the third key feature stated in line 180-183? Specifically, does the analysis demonstrate such an effect?

**Limitations:**

One unique challenge in decentralized bilevel optimization is the efficient computation of the upper level hyper-gradient, which requires the inverse of a large Hessian matrix. The method proposed in the paper requires a direct inversing operation done locally, which can limit the application when the local machine has limited computation power (e.g. in a federated learning setting). It would be great if a smarter decentralized algorithm can lift the requirement of directly inverting the Hessian matrix, while still enjoys the convergence guarantee.

**Strengths And Weaknesses:**

Strengths:
The paper is well-written as easy to understand. The considered decentralized bilevel optimization problem is an important topic. Several unique problems and theoretical challenges arising from the decentralization in a bilevel setting are addressed by the paper, and the theoretical result is also sound and convincing.

Weakness:
However, I do have several concerns: 1) Assumption A.1. (iv) requires the stochastic gradients to be bounded in expectation, which is stronger than the standard bounded variance assumption widely satisfied in practice, e.g. Assumption 3 in [1]; 2) In the proposed algorithm, computing the hyper-gradient for the upper level requires inverting a matrix $v_t^k$. The matrix has a size of $d_y \times d_y$, which is large if the network size is large. Therefore, inverting such a large matrix can be too computational intensive, especially for a local machine with limited computation power.

[1] Saeed Ghadimi and Mengdi Wang. Approximation methods for bilevel programming.

---

> ### Author Response · Authors · 2022-08-02
> **Responses to Reviewer 8YY4**
>
> Thanks for your extremely valuable comments, which have provided us a chance to clarify our assumptions and helped us improve our paper. Below we provide point-to-point responses to your comments.
>
> **Question.** Bounded second-moment assumption
> \
> **Response.** This is a very good point! First, for any stochastic gradient $\nabla_y f(x,y;\zeta)$, we can see  that
>      $\mathbf{E}\_\zeta[|| \nabla_y f(x,y;\zeta) ||^2] \leq 2 || \nabla_y f(x,y) ||^2 + 2 \mathbf{E}\_\zeta[|| \nabla_y f(x,y;\zeta)  - \nabla_y f(x,y) ||^2],$
>      which suggests that the bounded second moment can be implied by a bounded gradient plus a bounded variance assumption.
>      Second, it is worth emphasizing that the boundednesses of $\| \nabla_y f (x,y) \|$ and $\| \nabla\_{xy}^2 g(x,y)\|$ are necessary for ensuring the Lipschitz continuity of the overall gradient
>      $
>      \\nabla F(x) = \\nabla_x f(x,y) - \nabla_{xy}^2 g(x,y) [ \nabla_{yy}^2 g(x,y)]^{-1} \\nabla_y f(x,y).
>      $
>      Indeed, such assumptions are also adopted by   [1] (see Assumptions 1(b) and 2(e)).
>      Third, it is worth mentioning that our problem needs to consider the consensus errors while [1] does not need to. As this is the first paper in decentralized bilevel optimization, we focus on establishing a scheme to tackle this problem and assume the bounded second moment of $\| \nabla_x f(x,y;\zeta) \|$ to facilitate our analysis. In fact, for non-convex problems, we observe from Lemma B2 that the consensus errors decay to zero at the rate of $\mathcal{O}(1/T)$, which is much faster than the $\mathcal{O}(1/\sqrt{T})$ rate for the evaluation metric. Therefore,
>      We believe that this assumption might be relax-able to bounded variance. This is left for future work.
>
> [1] Saeed Ghadimi and Mengdi Wang. Approximation methods for bilevel programming.
>
> **Question.** Computation cost in inverting the hessian
> \
> **Response.**
> We agree that computing the inverse of hessian for large-scale problems is expensive. To overcome this issue, in our revised manuscript, we have refined our algorithm to approximate the inverse of hessian using $b = \Theta( \log(T))$ independent estimators $\\{ v_{t,j}^k \\}_{j=1}^b$of the hessian (lines 9-14 of Algorithm 1), which avoids such an expensive computation cost. Our refined algorithm still enjoys the $\mathcal{O}(\frac{1}{\sqrt{KT}})$ and $\mathcal{O}(\frac{1}{KT})$ convergence rates for non-convex and strongly convex problems, respectively.  The original approach suffers from a cost of $\mathcal{O}(d_y^3)$ in inverting the hessian within each iteration, while the per-iteration computation cost for our refined approach is $\mathcal{O}(\log(T) d_y^2)$, which is cheaper for high-dimensional problems.
>
> **Question.** Third key feature in lines 180-183
> \
> **Response.**
> We are sorry for the confusion caused by our previous argument. Indeed, the decentralized network is robust to contingencies in the network. If some communication channel fails, the agents can still jointly learn provided that the network is still connected, and the optimization objective remains unchanged. In contrast, a one-center-multi-agent system would become unconnected if the central server fails, which also leads to a change in the optimization objective.
>  We have revised our manuscript accordingly.

---

### Author Response · Authors · 2022-08-02
**Additional numerical experiments and more efficient approach for hessian inverse estimation**

We thank the reviewers for the suggestions about numerical experiments and computational cost in inverting the hessian. In our revision, we have conducted additional numerical experiments to test the performance of our algorithm over various network topologies and large-scale problem instances. We also
 develop a more efficient approach to estimate the inverse of hessian over decentralized networks, which enjoy the same convergence rate and lower computational cost. The details are as follows.

**Additional numerical experiments.**
 In our revision, we have provided the following additional experiments.
+ We test our algorithm over networks of different topologies, including a fully-connected network ($K=5,10,20,100$) and a randomly connected network ($K=5,10,20$). The additional experiments are reported in **Figures 4, 5, 6 and  Section E.1 of the revised supplement**. These results show that our algorithm converges rapidly under various network topologies and demonstrates its efficiency. Also, Figure 4 shows that our algorithm converges rapidly when $K = 100$, which demonstrates the efficiency of our algorithm for large networks.
+ We evaluate the average squared gradient norm of the solution path. We plot the logged squared grad norm $\log( \sum_{i=1}^t \|\nabla F(\bar x_i) \|^2 )$ against the log-iteration, $\log t$, in  Figure 6,7, Section E.1 of the supplement to validate our $\mathcal{O}(1/\sqrt{T})$  convergence rate result Theorem 1, with an additional  straight line of slope -0.5 provided for benchmark comparison.
Figure 6 suggests that the slope of $\log(\sum_{i=1}^t||\nabla F(\bar x_i)||^2)$ against $\log t$ is around $-1/2$, which validates our $\mathcal{O}(1/\sqrt{T})$ convergence rate claim in Theorem 1.

**More efficient approach for hessian inverse estimation.**
To reduce the computational cost of inverting the hessian, we have revised our algorithm to approximate the hessian inverse by using
 $b = \Theta( \log(T))$ independent estimators $\{ v_{t,j}^k \}_{j=1}^b$ of the hessian (lines 9-14 of Algorithm 1). This reduces the per-iteration computational cost from $\mathcal{O}(d_y^3)$ to  $\mathcal{O}(\log(T) d_y^2)$, making the algorithm tractable for high-dimensional problems.

---

### Meta-Review · Area_Chair_g9Mp · 2022-08-24

**Recommendation:** Accept
**Confidence:** Certain

**Metareview:**

This paper proposed a fully decentralized algorithm for bilevel optimization. Although the techniques are a combination of existing ones from bilevel literature and the decentralized optimization literature, but the setting considered is considerably sophisticated (i.e., both levels are distributed). The algorithm is a single-timescale, and the rates are good for both nonconvex and convex settings. The reviewers all appreciate the contribution of this work. Therefore I recommend acceptance of the paper.

**Award:**

No

---

### Decision · Program_Chairs · 2022-09-14

Accept